# Neuroprotective gap-junction-mediated bystander transformations in the adult zebrafish spinal cord after injury

Andrea Pedroni[1,2], Yu-Wen E. Dai[1,2], Leslie Lafouasse [1], Weipang Chang[1], Ipsit Srivastava[1], Lisa Del Vecchio[1] & Konstantinos Ampatzis [1] ✉

The adult zebrafish spinal cord displays an impressive innate ability to regenerate after traumatic insults, yet the underlying adaptive cellular mechanisms remain elusive. Here, we show that while the cellular and tissue responses after injury are largely conserved among vertebrates, the large-size fast spinal zebrafish motoneurons are remarkably resilient by remaining viable and functional. We also reveal the dynamic changes in motoneuron glutamatergic input, excitability, and calcium signaling, and we underscore the critical role of calretinin (CR) in binding and buffering the intracellular calcium after injury. Importantly, we demonstrate the presence and the dynamics of a neuron-to-neuron bystander neuroprotective biochemical cooperation mediated through gap junction channels. Our findings support a model in which the intimate and dynamic interplay between glutamate signaling, calcium buffering, gap junction channels, and intercellular cooperation upholds cell survival and promotes the initiation of regeneration.

The primary mechanical insult in spinal cord injury (SCI; primary injury) initiates a progressive state of neuronal degeneration (secondary injury) that amplifies the damage to adjacent intact tissue[1–3]. While the precise molecular and cellular pathways of SCI-induced degeneration remain elusive[4], glutamate-mediated excitotoxicity is well recognized as one of the principal contributors[3–11]. The excessive release of the excitatory transmitter glutamate during the early stages of secondary injury leads to significant dysregulation of intracellular calcium homeostasis[12], in which calcium ($Ca^{2+}$) plays an essential role in initiating a series of downstream cellular processes that drive cell death[3,11,13]. Moreover, calcium ions and their succeeding cytotoxic death factors can exert their effects on adjacent cells through gap junction channels by the so-called bystander cell-killing effect[14,15]. Accordingly, transient up-regulation of connexins (Cx; the proteins forming gap junctions) and gap junction channels follow a wide range of nervous system insults[16–19], including glutamate-mediated excitotoxicity[10,20,21].

Most neurons possess calcium-binding proteins as a mechanism to counteract disturbances of calcium homeostasis by binding and buffering the augmented calcium[22–26]. However, the mammalian large fast-fatigable MNs (FF-MNs) are selectively vulnerable to calcium-dependent glutamate excitotoxicity[8,9,27–29] due to inadequate expression or complete absence of calcium-binding proteins (30-324) hence, they remain unprotected to drastic alterations in $Ca^{2+}$ concentration.

In contrast to mammals, adult zebrafish have an impressively high innate regenerative capacity of the spinal cord after injury[30–32]. It is still unknown whether the SCI in adult zebrafish can cause fast MN vulnerability similar to that of mammals. Additionally, it is unclear whether the zebrafish motoneurons possess any protective mechanisms against potential cytotoxic stimuli. Thus, determining the adaptive cellular mechanisms that occur in the zebrafish adult fast primary motoneurons after injury was of great interest. We also reasoned that specifying the temporal repertoire and distribution pattern of gap junction expression in motoneurons after injury would provide insights into how intercellular communication might modulate cell viability and influence the regeneration process.

[1]Department of Neuroscience, Karolinska Institutet, 171 77 Stockholm, Sweden. [2]These authors contributed equally: Andrea Pedroni, Yu-Wen E. Dai. ✉e-mail: Konstantinos.Ampatzis@ki.se

## Results

The large-size, fast-fatigable mammalian spinal MNs (FF-MNs) are particularly vulnerable to neurodegenerative conditions, including injury[7,8,27–29,33]. The zebrafish motor column contains a small number (~4/hemisegment) of oversized early-born motoneurons (Fig. 1a–c and Supplementary Fig. 1), called primary MNs (pMNs), which innervate the fast muscles and are recruited exclusively in fast movements[34–38]. Therefore, the possibility arises that, similarly to mammals, spinal cord injury (SCI) mediated degeneration may predominantly affect the zebrafish fast pMN pool. To test this possibility, we assessed the presence of retrogradely labeled pMNs caudally (segments 13 and 14) to the lesion site (lesion at segment 12) at 1, 3, and 7 days after the complete transection of the spinal cord (Fig. 1d). Our analysis revealed that all the caudally to injury located pMNs remain in place after injury (Fig. 1e). Next, we examined potential changes in pMN soma size as neuronal cell death is often manifested by distinct morphological criteria such as soma shrinkage or swelling[39]. We did not observe any significant changes in the soma sizes of the pMNs after the injury or during the regeneration process, suggesting that the pMNs did not undergo cell death-associated morphological pressure (Fig. 1e). To confirm the pMN viability after injury, spinal cords were treated with propidium iodide (PI), which labels dead cell nuclei (Fig. 1f). The vast majority of PI⁺ cells 24 h after injury were detected close to the lesion site, with an overall significant reduction of dead cell number at 3- and

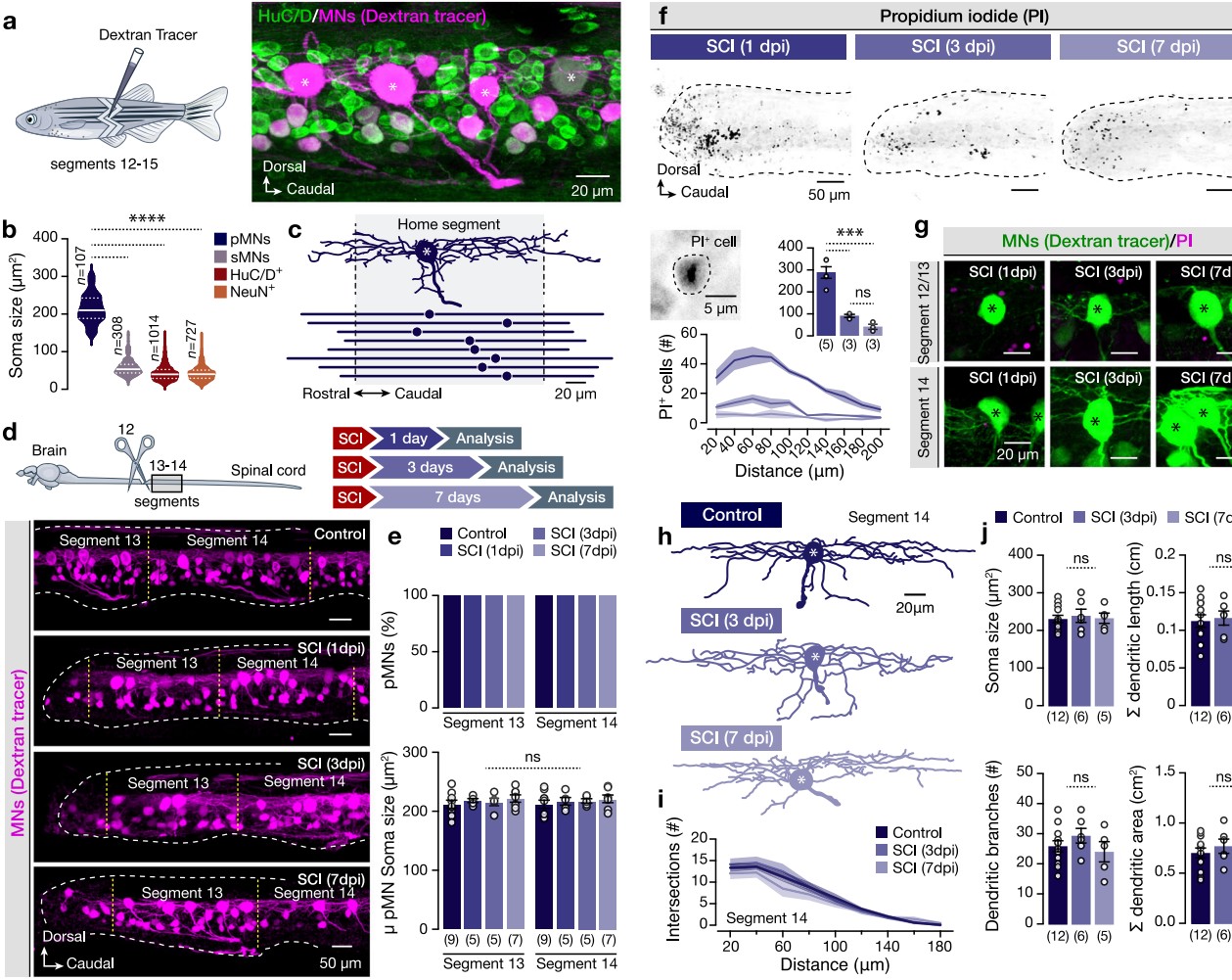

**Fig. 1 | Primary MNs, the largest spinal cord neurons, survive after injury.**
**a** Experimental approach for retrograde labeling of spinal motoneurons (MNs) by injecting the Dextran tracer. Lateral view of whole-mount confocal fluorescence image showing traced MNs (magenta) along with all spinal neurons (HuC/D⁺, green). **b** Quantification of the soma size of motoneuron types (pMNs and sMNs) and other spinal neurons (HuC/D⁺ and NeuN⁺, excluding the dextran traced pMNs). **c** Reconstructed neurobiotin-filled pMN showing the extended dendritic arborizations to adjacent spinal cord segments. The dashed black lines define the home segment (shaded area). Illustration of the pMN soma (solid circle) and the rostrocaudal distribution of the dendrites obtained from the lateral view reconstructions. The lines represent the maximum and minimum rostrocaudal dendritic position. **d** Experimental design and representative whole-mount microphotographs show retrogradely traced motoneurons (magenta) in control, after injury, and during regeneration. Yellow dashed lines indicate the spinal segments. **e** Analysis of the proportion of the retrogradely traced pMNs number per hemisegment located caudally to the injured site. Quantification of the pMN soma size in

all experimental conditions. **f** Inverted confocal images from whole-mount adult zebrafish spinal cord showing the PI⁺ cells after injury. Spatial distribution and quantification of PI incorporating cells (PI⁺) in all experimental conditions.
**g** Retrogradely traced pMNs (green), do not incorporate PI (magenta) at 1 day (n = 5 zebrafish), 3 days (n = 3 zebrafish) and 7 days (n = 3 zebrafish) post-injury.
**h** Reconstructed examples of neurobiotin-filled pMNs of segment 14 in different experimental conditions. **i** Sholl analysis of dendritic complexity with a decreasing radial distance from the pMN soma (epicenter). **j** Analysis of major dendritic tree properties of pMNs shows no significant morphological changes after injury. Asterisks indicate the pMN cell bodies. μ population mean, Σ summation (total), dpi days post-injury, HuC/D elav3 + 4, NeuN neuronal protein (Fox-3/Rbfox3), PI propidium iodide, pMN primary motoneuron, SCI spinal cord injury, sMN secondary motoneuron. Data are presented as mean ± s.e.m., and as violin plots. ***P < 0.001; ****P < 0.0001; ns, not significant. For detailed statistics, see Supplementary Table 1. Source data are provided as a Source Data file.

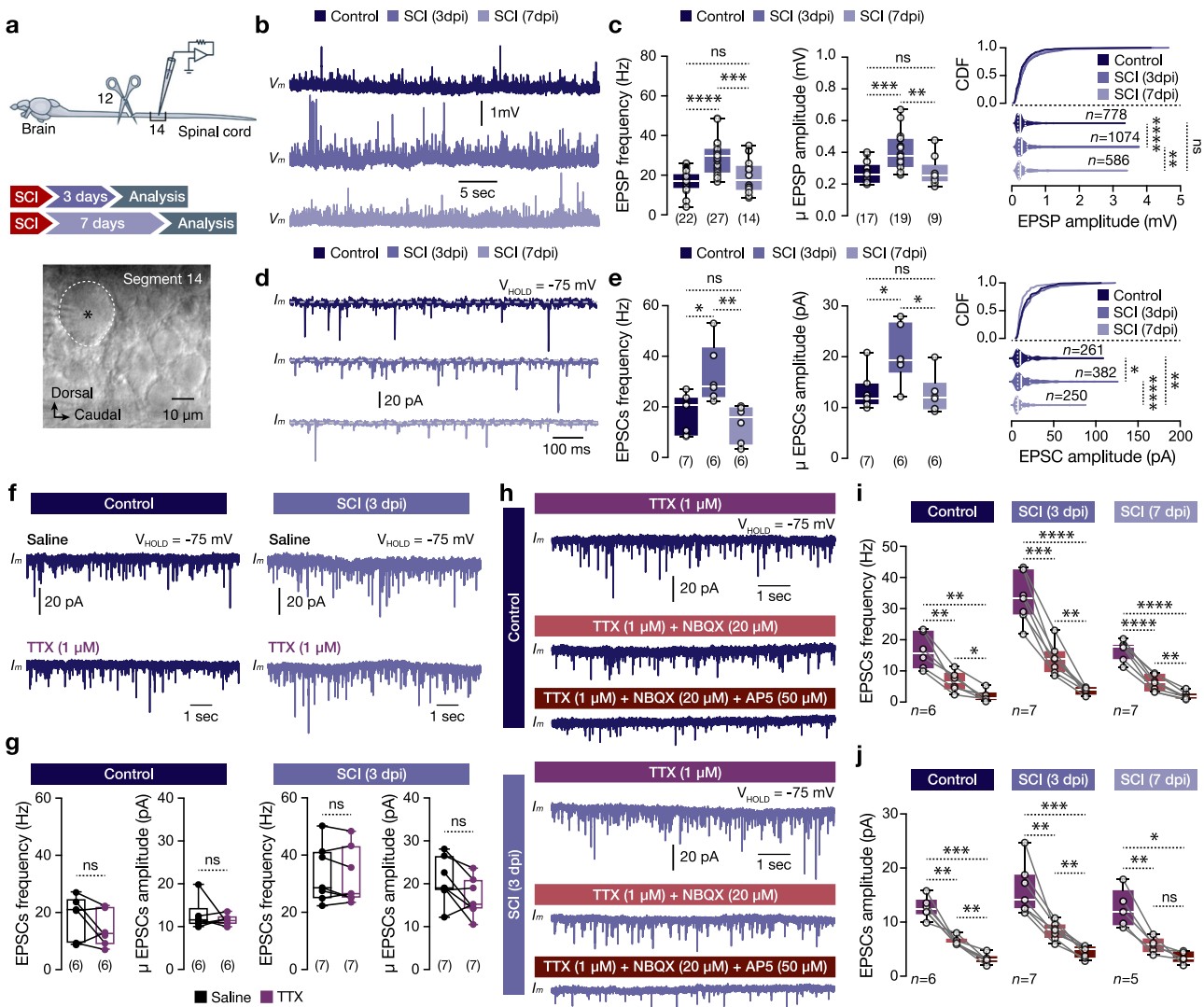

**Fig. 2 | Changes in synaptic input to pMNs after injury. a** Ex vivo setup of isolated spinal cord preparation allows whole-cell patch-clamp recordings of pMNs. The dashed black line indicates the recorded cell (pMN). **b** Representative traces of the recorded spontaneous current-clamp excitatory post-synaptic potentials (EPSPs). **c** Analyses of the recorded EPSP frequency and amplitude in all experimental conditions. **d** Representative pMN voltage-clamp recordings showing spontaneous post-synaptic events (EPSCs) recorded in control and after injury (SCI). **e** Analyses of the EPSCs frequency and amplitude in different experimental conditions. **f** Sample voltage-clamp traces obtained from pMN in saline and after bath application of TTX. **g** Analysis of the recorded EPSCs (frequency and amplitude) shows no changes after pharmacological application of TTX. **h** Exogenous bath application of TTX, followed by the glutamate receptor antagonists NBQX and AP5,

reduced the frequency of the recorded excitatory post-synaptic events (EPSCs). **i** Analysis of the EPSC frequency (Hz) changes after sequential blockage of glutamate receptors in all experimental conditions. **j** EPSC amplitude quantification and analysis after sequential application of the glutamate receptor antagonists. The asterisk indicates the pMN cell body. µ population mean, AP5 D-(-)-2-Amino-5-phosphonopentanoic acid, CDF cumulative distribution (frequency), dpi days post-injury, EPSC excitatory postsynaptic current, EPSP excitatory postsynaptic potential, NBQX 2,3-dioxo-6-nitro-7-sulfamoyl-benzo[f]quinoxaline, SCI spinal cord injury, TTX tetrodotoxin. Data are presented as box plots showing the median with 25/75 percentile (box and line) and minimum–maximum (whiskers). *$P < 0.05$; **$P < 0.01$; ***$P < 0.001$; ****$P < 0.0001$; ns, not significant. For detailed statistics, see Supplementary Table 1. Source data are provided as a Source Data file.

7-days post-injury (dpi) (Fig. 1f). However, none of the pMNs or secondary MNs (sMNs; Supplementary Fig. 2) in proximity and more distal to the lesion site in all tested conditions were found to incorporate PI into their nuclei (Fig. 1g and Supplementary Fig. 2). Based on the extensive pMN dendritic spread (Fig. 1c and Supplementary Fig. 1c, d), we expected that SCI would potentially damage the dendrites of the proximal pMNs (segments 13), whereas the more distal (segments 14) would be physically unaffected. Hence, we focused on pMNs of segment 14 (undamaged) and sought to determine potential changes in their morphology. Primary MNs were filled intracellularly with neurobiotin (NB) and subsequently reconstructed using confocal microscopy (Fig. 1h). The overall dendritic morphology of the pMNs remained unaltered after the injury and during the regeneration phase

(Fig. 1i, j). Altogether, our results argue against the notion of the selective vulnerability of the large-size fast pMNs, or the vulnerability of any MN class in the adult zebrafish spinal cord prompting us to assess whether spinal cord injury could trigger an excessive release of glutamate, as reported before in mammals[3–6,8,9,11].

To determine whether spinal cord injury could initiate an aberrant release of glutamate to undamaged pMNs in the adult zebrafish spinal cord, we performed visually guided whole-cell electrophysiological recordings from the pMNs located in segment 14 (physically undamaged; Fig. 2a). We observed a significant increase in the frequency and amplitude of recorded excitatory post-synaptic events (potentials, EPSPs; currents, EPSCs) at 3 days post-injury (Fig. 2b–e), whereas 7 days of recovery from the spinal cord injury restored the frequency of the

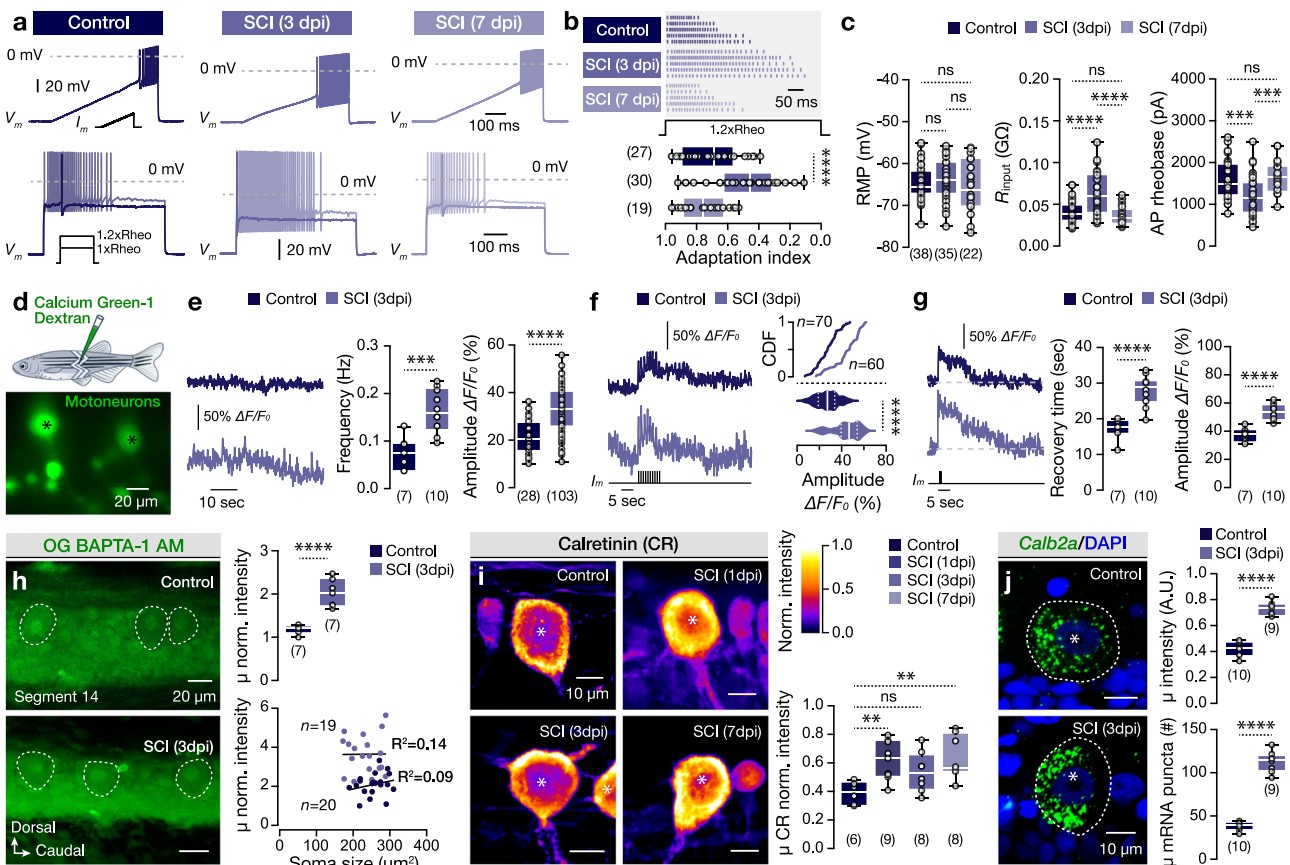

**Fig. 3 | Electrical, cellular, and biochemical changes in the pMN population after injury. a** Repetitive firing pattern in response to depolarizing ramp-induced and current steps-induced firing. The solid color traces show the pMN response at the rheobase and faint traces at 120% of the rheobase. **b** Representative sample of 5 pMNs per experimental group and analysis of the firing adaptation index. **c** The resting membrane potential (RMP), input resistance (*R*input), and rheobase of the recorded pMNs. **d** Functional calcium imaging of retrogradely traced spinal motoneurons (green) with the calcium indicator Green-1 Dextran. **e** Representative traces of pMN spontaneous activity in control and after injury (3 dpi). Quantification and analysis of the recorded events' frequency and amplitude. **f** Fluorometric recordings and analysis of calcium transients' amplitude following a local electrical stimulation of the spinal cord with 10 pulses (1 Hz). **g** Functional calcium imaging traces after a local electrical stimulation of the spinal cord with 15 pulses (30 Hz), and quantification of the calcium transient amplitude and recovery time. **h** Fluorescent images (single-plane) after loading with cell-permeant calcium-

sensitive dye Oregon Green BAPTA-1 AM. Quantification of the resting signal intensity of the pMNs. Primary MN resting calcium intensity is independent of the soma size. **i** Normalized intensity and analysis of immunolabelled pMNs with calcium-binding proteins CR from uninjured (control) and injured animals. Normalizations of the arbitrary units were performed for CR calcium-binding protein to the highest obtained intensity value. **j** *Calb2a* mRNA expression (Green) in pMNs (DAPI; Blue). Quantification and analysis of the average normalized intensity and puncta of *Calb2a* in pMNs per animal. Asterisks and dashed white lines indicate the pMN cell bodies. AP action potential, CDF cumulative distribution (frequency), CR calretinin, DAPI 4′,6-diamidino-2-phenylindole (nuclear staining), dpi days post-injury, Rheo rheobase, *R*input input resistance, RMP resting membrane potential, SCI spinal cord injury. Data are presented as violin plots and as box plots showing the median with 25/75 percentile (box and line) and minimum–maximum (whiskers). \*\**P* < 0.01; \*\*\**P* < 0.001; \*\*\*\**P* < 0.0001; ns, not significant. For detailed statistics, see Supplementary Table 1. Source data are provided as a Source Data file.

excitatory events to that in control animals, suggesting that the observed increase of excitatory drive is dynamic and reversible (Fig. 2b–e). Moreover, blocking the ability of all spinal cord neurons to fire action potentials with tetrodotoxin (TTX; Supplementary Fig. 3) resulted in no change in the frequency or amplitude of EPSCs (Fig. 2f, g), confirming that the recorded events are not action potential mediated. To gain insight into the precise nature of the excitatory inputs received by the pMNs, we applied the selective antagonist of AMPA and Kainate glutamate receptors (NBQX), followed by the NMDA receptor antagonist (AP5) (Fig. 2h). NBQX reduced the EPSCs frequency by ~60%, and further reduction (~29%) was observed after applying AP5 (Fig. 2i), confirming that ~90% of the recorded excitatory inputs are indeed glutamatergic. For all tested conditions and EPSCs amplitude, similar findings were obtained, suggesting that SCI does not change glutamate receptor relative abundance on pMNs (Fig. 2j and Supplementary Fig. 4). Collectively, our results confirm that at 3 days post-injury, the undamaged pMNs experience a transient wave of glutamatergic drive, similar to what was reported previously in the mammalian spinal cord.[4]

To ascertain whether glutamate release also affects the excitability of the undamaged pMNs close to the injury site, the repetitive firing of pMNs (segment 14) was evaluated in response to increased depolarizing current injection protocols (Fig. 3a). Following injury (3 dpi) pMNs generated more action potentials (Fig. 3a) and they displayed a significantly reduced adaptation index (Fig. 3b), an indicator of neuronal excitability. Further analysis of several electrical properties revealed dynamic adaptations in pMNs during the time course of injury (Fig. 3c and Supplementary Fig. 5). Notably, the increased input resistance (*R*input) and decreased rheobase at 3 dpi without changes in the resting membrane potential (RMP), confirmed the observed post-injury-induced pMN excitability (Fig. 3c and Supplementary Fig. 5a). We next tested whether the increased glutamate release and excitability could affect the pMN intracellular calcium concentrations. Fluorometric recordings were performed using the long-wavelength calcium indicator Calcium Green-1 Dextran (3000 MW) by injecting the dye into the myotomes to achieve stochastic retrograde labeling of spinal MNs (Fig. 3d). All pMNs displayed an increased baseline

fluorometric activity after injury (at 3 dpi; Fig. 3e). Upon electrical stimulation of the spinal cord (segment 14) with 10 pulses (at 1 Hz), the evoked transients in injured animals (3 dpi) were significantly larger compared to those of the uninjured animals (Fig. 3f). Moreover, stronger stimulation of 15 pulses (at 30 Hz) revealed that in addition to the increased amplitude of the evoked calcium transients, the recovery time to baseline was also increased by 65% after injury (Fig. 3g). To further determine the resting (or basal) intracellular calcium in pMNs, spinal cords were treated with the cell-permeant calcium-sensitive dye Oregon green 488 BAPTA-1 AM, for an acute network loading, revealing an overall increase of the resting calcium in the adult zebrafish spinal neurons (Supplementary Fig. 6), with the pMNs to display a mean 74% increase at 3 dpi that was not correlated to their soma size (Fig. 3h). Together these results suggest that pMNs exhibit increased resting and uptake or release of cytosolic calcium after injury.

The increased excitatory input and activation of the glutamate receptors after injury are expected to increase the cytosolic calcium to unwanted levels[13,23,25,40]. As a defense mechanism, neurons employ calcium-binding proteins to bind and buffer excess calcium. Accordingly, the adult zebrafish spinal pMNs retain the ability to express two principal calcium-binding proteins, Parvalbumin (PV) and Calretinin (CR, 29-kD)[41]. To determine whether adaptations in the expression of calcium-binding proteins occur after injury, the intensity of Parvalbumin (PV) and Calretinin (CR) proteins was evaluated in the pMN somata (Fig. 3i and Supplementary Fig. 7). While the normalized intensity of the PV expression remained unchanged in pMNs (Supplementary Fig. 7), a rapid (at 1dpi) upregulation of the CR was detected (Fig. 3i). Interestingly, at 3 dpi, we observed a reduction in the CR intensity levels (Fig. 3i), while the increased production of CR was maintained by the increased detection of *Calb2a* mRNAs (Fig. 3j). These observations indicate the potential overuse of the "free" (unsaturated) CR protein (as seen from immunodetections in Fig. 3i) to buffer the augmented intracellular calcium, as the $Ca^{2+}$-saturated calretinin undergoes significant re-structural and conformational changes (cleavage)[42,43]. In addition, we observed a significant 50% increase in the number of $CR^+$ neurons in the adult zebrafish spinal cord after an injury that displayed similar topographic organization and soma sizes (Supplementary Fig. 8), and in all cases, the pMNs were among the spinal neurons with the highest intensity of CR expression (Supplementary Fig. 8d). Although, we did not detect any changes in the CR expression pattern between the different motoneuron pools[41], we found a significant increase among the fast secondary MN pool (Supplementary Fig. 8e-g). Our findings establish Calretinin (CR) as a critical molecule involved in post-injury-induced cellular events and potentially crucial for neuroprotective and restorative mechanisms. These findings support a previous study that linked the expression of CR to the regeneration process of the adult zebrafish spinal cord[44].

Previous studies highlighted an essential link between injury and gap junction-mediated intercellular communication, showing the direct spread of potential cytotoxic substrates between neurons after traumatic insults, which amplifies the damage, known as "bystander effect"[16–19,21]. Between the different zebrafish spinal cord neurons, the somata and the proximal neurites of the pMNs were heavily decorated by connexins 35/36, the primary proteins forming gap junction channels in neurons[10,15,18,19,21,45–47] (Fig. 4a and Supplementary Fig. 9). To map possible changes in the expression of the connexin 35/36 in the pMN pool after the injury, we evaluated the normalized immunostaining intensity and the number of antibody aggregations (puncta) on the pMN cell somata (Fig. 4b, c). We identified a dynamic upregulation of the connexin 35/36 expression shortly after the injury (at 1dpi) with a peak of the expression at 3 days post-injury (Fig. 4b, c). Partial recovery from injury (7dpi) significantly reduced the levels of connexins 35/36 expression to levels below the ones of the control (uninjured) animals (Fig. 4b, c), suggesting a homeostatic regulation of gap junction protein expression and assembly.

Next, we sought to determine whether the observed homeostatic regulation of gap junction protein expression in pMNs is functional using dual whole-cell electrophysiological recordings (Fig. 4d). Dual recordings between pMNs of the same segment confirmed a reliable (100%) bidirectional electrical coupling between all pMNs with a coupling coefficient that was not dependent on the distance between the neurons within the spinal cord segment nor their relative rostrocaudal position (Fig. 4e and Supplementary Fig. 10). Our analysis also revealed a significant dynamic increase in electrical coupling between pMNs at 3 days post-injury that was further reduced at 7 days post-injury (Fig. 4f), comparable to the expression of connexin proteins (Fig. 4b, c).

The increased expression of connexins affected the electrical connectivity of pMN neurons, which are always interconnected (Fig. 4e and Supplementary Fig. 10). However, it is unclear whether gapjunction channels play a role in reconfiguring the spinal cord network and affecting the interconnectivity between previously uncoupled neurons (dye coupling). To test this possibility, an injection of Neurobiotin (NB, Biotin ethylenediamine, 287 Da), a low molecular weight tracer that passes across gap junction channels[46,48,49], was performed from a single identified pMN (Fig. 4g). Our analysis showed that most dye-coupled neurons were located rostrally to the NB-filled pMN (Fig. 4g, h). Quantification of dye coupling showed a significant increase (50%) in the NB-labeled cells at 3 dpi (Fig. 4g–j) that were also in close proximity to pMN (radius of <30 μm; Fig. 4i), while at 7 dpi, the population and the spatial distribution of the dye-coupled cells returned to the control levels (Fig. 4g–j), suggesting a dynamic regulation in the interconnected neurons, similar to our previous results on the expression of the connexins 35/36. Moreover, there was no difference in the sizes of the dye-coupled cells in all conditions (Fig. 4j). Yet, it remained uncertain whether the rise in dye coupling is a result of the gap-junction connections between neurons or if the connexins function as hemichannels that facilitate the diffuse release and uptake of NB through the cell membrane and the extracellular space. To gain insight into this possibility, we evaluated the specificity of dye coupling by combining the NB tracing with immunodetection of the neurons using the pan-neuronal marker HuC/D (Supplementary Fig. 11a, b). We observed selectivity in dye coupling (Supplementary Fig. 11a) and a close relation between the pMN cellular structures (soma, axon, dendrites) to the dye-coupled neuron soma (Supplementary Fig. 11c, d). These results do not support the role of neuronal hemichannels in intercellular reconfiguration and communication after injury, supporting previous studies[50]. We also conducted pair recordings between a pMN and unidentified spinal neurons within the same segment to determine if the observed increase in dye-coupling results in functional connections between neurons (Fig. 4k). Our findings showed that the probability of connectivity increased from 12% to 22.7% after injury, especially between neurons that were closer to pMN (Fig. 4l). Additionally, we observed that after the injury, the coupling coefficient was significantly larger towards the pMNs than from pMNs (Fig. 4m). Our data suggest that increasing connexin protein expression after the injury is crucial for reconfiguring spinal cord networks and altering connectivity strength; however, the cellular mechanism that drives these essential adaptations remained unclear.

Based on the observed increase in the glutamatergic input to pMNs (Fig. 2), we asked whether glutamate impinges the gap junction channel protein expression. To address this question, we quantified the Cx35/36 immunolabeling after a single administration of NMDA and AMPA agonists of the ionotropic glutamate receptors (iGluRs) or the LY379268, a selective and highly potent agonist of the metabotropic glutamate receptors 2 and 3 (mGluR2/3) (Fig. 5). The expression of Cx35/36 was increased significantly 24 h after the administration of the group II mGluRs (mGluR2/3) agonist (Fig. 5a, b), as observed before in rodents[18,19] and was manifested by changes in the coupling coefficient (Fig. 5c). Consistent with these results, intraperitoneal administration of the mGluR2/3 agonist (LY379268) or antagonist (LY341495)

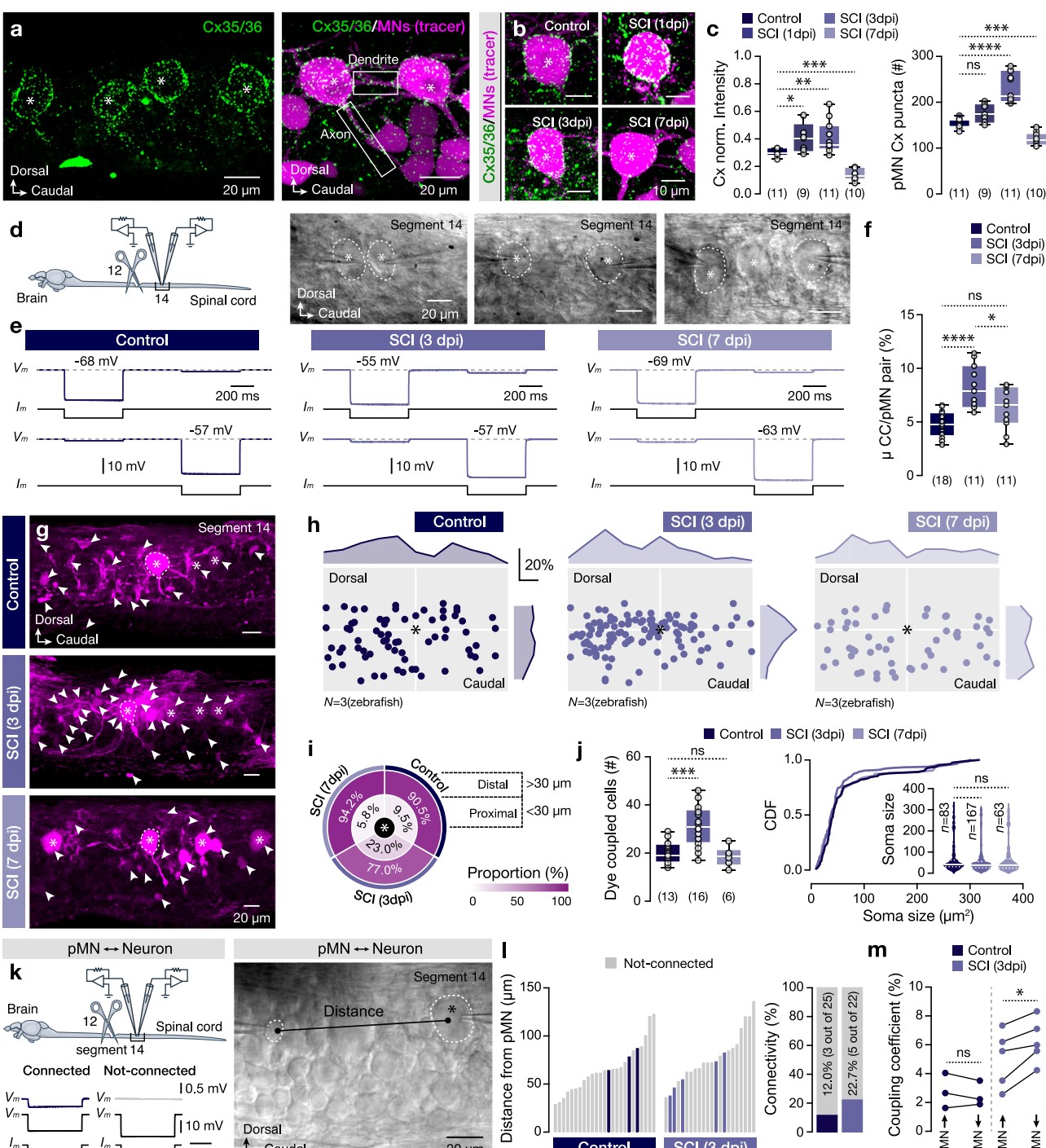

**Fig. 4 | Post-injury changes in pMN connexin proteins abundance affect the electrical and dye coupling. a** Whole-mount image shows the expression of Cx35/36 (green) in the pMNs (asterisks). The expression of Cx35/36 (green) in proximal neurites of the retrogradely traced pMN (magenta; *n* = 11 zebrafish).
**b** Representative immunofluorescent images show the Cx35/36 expression in traced pMN somata in different experimental conditions. **c** The Cx35/36 normalized intensity and the number of aggregations (puncta) in the pMN somata. **d** Ex vivo setup that allows dual recordings between pMNs (asterisks). The white dashed line indicates the recorded pMNs. **e** Sample average traces (~40 sweeps) show the bidirectional electrical coupling between pMNs. **f** Mean coupling coefficient (CC) between two pMNs. **g** Representative confocal images showing the dye coupling (NB⁺-cells, magenta) after intracellular loading of a single pMN (white dashed line) with NB in control (*n* = 13 zebrafish), 3 dpi (*n* = 16 zebrafish) and 7 dpi (*n* = 6 zebrafish). Arrowheads indicate the NB⁺ neurons. **h** Representative spatial distribution of dye-coupled neurons (colored filled circles) from a pMN (centered asterisk) in

the adult zebrafish spinal cord (segment 14). Data are presented as a group of 3 animals per experimental condition. **i** Proximity analysis of the dye-coupled cells (NB⁺). **j** Quantification of the number and soma size of the dye-coupled neurons (NB⁺). **k** Ex vivo setup for performing pair recordings between a pMN and other spinal neurons (unknown) to define the electrical coupling probability (connected/not-connected). Image showing a pair of connected neurons (pMN and spinal neuron) and the definition of their Euclidean distance. The white dashed line indicates the recorded neurons. **l** Analysis of the distance and probability of electrical connectivity (%) from a pMN. **m** Coupling coefficient (in %) analysis. Asterisks indicate the pMN cell bodies. μ population mean, CC coupling coefficient, Cx connexin, dpi days post-injury, pMN primary motoneuron, SCI spinal cord injury. Data are presented as violin plots and box plots showing the median with 25/75 percentile (box and line) and minimum–maximum (whiskers). *P < 0.05; **P < 0.01; ***P < 0.001; ****P < 0.0001; ns, not significant. For detailed statistics, see Supplementary Table 1. Source data are provided as a Source Data file.

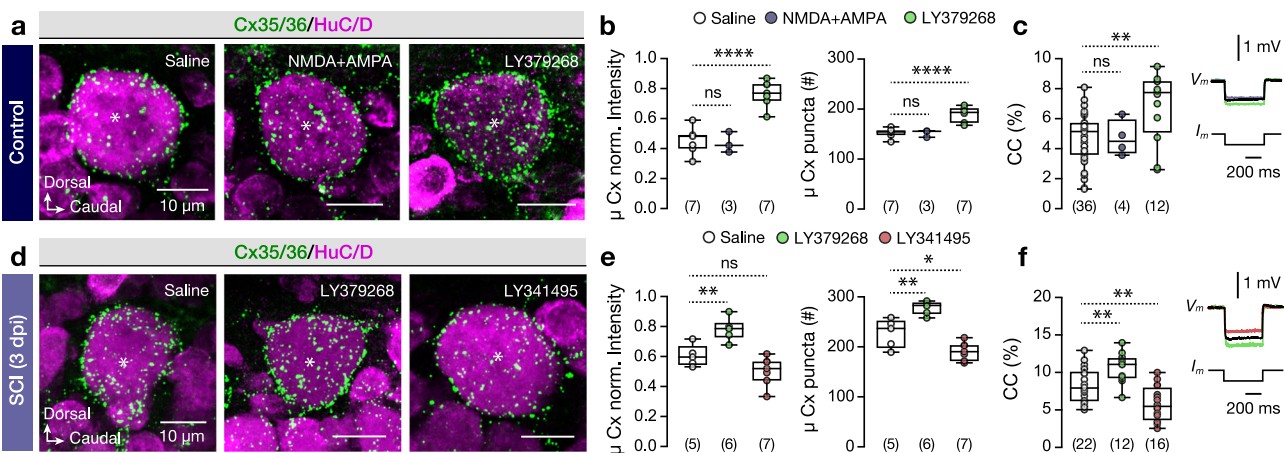

**Fig. 5 | mGluR2/3 receptors are crucial for the abundance of connexin proteins in pMNs. a** Representative immunofluorescent images of the Cx35/36 (green) expression on adult spinal pMNs somata (HuC/D⁺; magenta) following the administration of the ionotropic glutamate receptor agonists (NMDA and AMPA) or after the selective agonist of the metabotropic mGluR2/3 (LY379268) in control (uninjured) animals. **b** Quantification and analysis of the Cx35/36 normalized intensity and the number of aggregations (puncta) in the pMN somata in all experimental conditions. **c** Analysis of the coupling coefficient (CC; as a percentage) between two pMNs in all conditions followed by superimposed color-coded sample traces showing the differential responses in hyperpolarizing current step after pharmacological administration of the ionotropic and metabotropic glutamate receptors. **d** Confocal images show pMN somata (HuC/D⁺; magenta) and the Cx35/36 (green) expression after the intraperitoneal injection of a selective mGluR2/3 agonist (LY379268) or antagonist (LY341495) at 3 dpi. Source data are provided as a Source Data file. **e** Analysis of the Cx35/36 normalized intensity and the number of puncta in the pMN somata in all conditions. **f** Analysis of the coupling coefficient (CC; as a percentage) between two pMNs at 3dpi in all conditions. Superimposed color-coded traces showing the differential responses in the hyperpolarizing current step after pharmacological administration of the metabotropic group II mGluRs agonist and antagonist. Asterisks indicate the pMN cell bodies. μ population mean, AMPA α-amino-3-hydroxy-5-methyl-4-isoxazole propionic acid, CC coupling coefficient, Cx connexin, dpi days post-injury, HuC/D elav3 + 4, NMDA N-methyl-D-aspartate, SCI spinal cord injury. Data are presented as box plots showing the median with 25/75 percentile (box and line) and minimum–maximum (whiskers). *P < 0.05; **P < 0.01; ****P < 0.0001; ns, not significant. For detailed statistics, see Supplementary Table 1. Source data are provided as a Source Data file.

after injury (3 dpi) significantly increased or decreased the expression of the Cx35/36 and the coupling coefficient respectively (Fig. 5d–f), suggesting a causal link between post-traumatic glutamate release and gap junction-mediated adaptations in spinal cord intercellular communication.

While gap junction channels are not only essential for several neuronal network functions[45–47], they also offer a direct cytoplasmic continuity between neurons, thus mediating biochemical communication by allowing the exchange of signaling molecules, metabolites, and ions up to 1.5 kDa[10,13,15,19]. Motivated by our previous findings on the upregulation of the calcium-binding protein Calretinin after injury, we hypothesized that the post-traumatic increase of the dye coupling and interconnectivity serves as a possible calcium buffering mechanism to counteract disturbances of the injury-induced Ca²⁺ homeostasis. To test this hypothesis, we first examined whether the dye-coupled neurons retain the ability to produce Calretinin. We observed that the vast majority (~75%) of the neurons interconnected through gap junction channels to a pMN are CR⁻ (Fig. 6a, b). Following the spinal cord injury (3 dpi), we observed a significant increase (66%) in the number of NB⁺CR⁻ neurons that also had different soma sizes (Fig. 6a–d). To further test whether the pMN calretinin binds and buffers Ca²⁺ diffused through the gap junction channels, we evaluated the intensity levels of the calretinin expression after applying the non-selective gap junction blocker carbenoxolone (CBX;[51]) in a dose that could considerably block the electrical and dye coupling without producing any severe cytotoxic events such as cell death (Supplementary Fig. 12). We detected a significant increase in the intensity of the CR levels after the administration of CBX (Fig. 6e). In addition, administration of CBX in injured animals revealed a rescue (as shown in Fig. 3i) of the CR intensity in 3 dpi pMNs (Fig. 6f) to similarly high levels detected at 1 and 7 dpi (Fig. 3i). These data postulate that the pMN calretinin could bind and buffer diffused calcium ions from neurons that are potentially incapable of this process.

Despite the overall coherence of our results, it is challenging to make sweeping generalizations about the crucial function of gap

junction channels in transforming the organization and operation of the spinal network because our study was limited to a single type of neuron, namely the pMNs. To determine if this adaptation mechanism reported here is highly contextual to pMNs or if our findings could be generalized, we injected animals with a mixture of Dextran tracer and neurobiotin (NB) into the axial musculature and ventral roots (Fig. 7a). Following this strategy, all axial MNs (pMNs and sMNs) were labeled with both markers (Dextran⁺NB⁺) and all the dye coupled spinal neurons to MNs were labeled only with neurobiotin (NB⁺; Fig. 7a). We observed a significant increase in the number and size of dye-coupled neurons to the spinal motor neuron (MN) population after injury (Fig. 7b, c). Part of these neurons had an interneuron identity (V2a-interneurons; Supplementary Fig. 13), supporting the partial generalization of our findings. Next, to test whether the increased bystander intercellular connection may in part contribute to cell viability and survival, animals injected with CBX after injury and the cell death was evaluated at 3dpi (Fig. 7d). A significant increase of PI⁺ cells (176%) was detected with a notably expanded distribution to more distal areas from the injury site (Fig. 7d). We also noticed a significant increase of PI⁺ cells expressing the pan-neuronal marker HuC/D, with small to medium soma sizes and wider extended distribution (Fig. 7e, f). Then, we asked whether the blockade of gap junction-mediated neuroprotection could affect the regeneration process. To test this possibility, the bridge formation progress (regeneration index; Fig. 7g) was compared between CBX- and Saline-injected animals (Fig. 7h). We detected a delay of spinal cord bridging with a significant decrease in the regeneration progress at 7 and 14 dpi (Fig. 7h, i). Finally, we tested whether the delay in the spinal cord regeneration process has an impact on the number of neurites crossing the lesion site (Fig. 7j). While we found a trend in CBX-treated animals to detect fewer retrogradely labeled processes that cross the injury site, it was not statistically significant at the cutoff level (P = 0.05; Fig. 7j). Altogether, these data are consistent with our interpretation that gap junction channels mediate essential post-injury roles in the adult zebrafish spinal cord and during the initial regeneration phase.

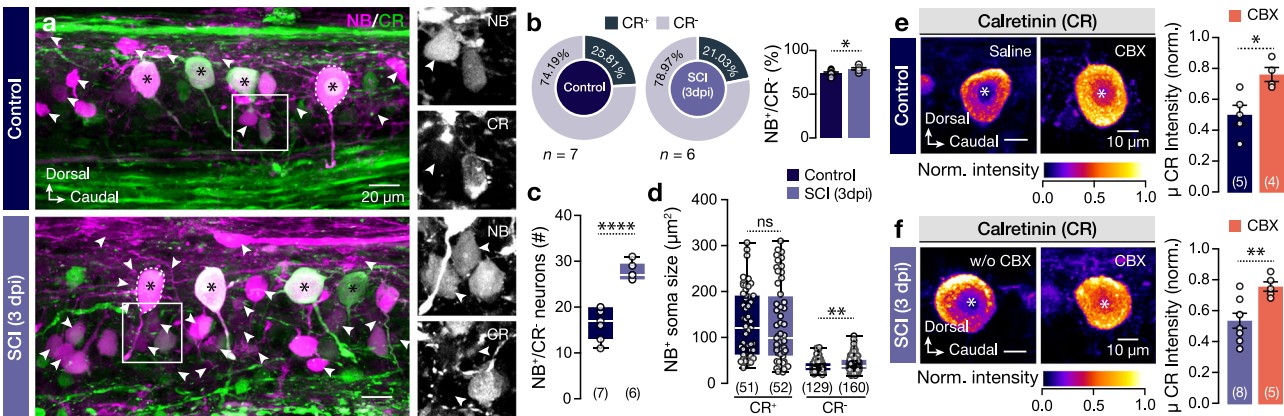

**Fig. 6 | Increase of dye-coupled neurons that lack calretinin after injury.**
**a** Representative stack of confocal fluorescence images from whole-mount pre-parations showing the dye-coupled cells (NB+, magenta) from a pMN (white dashed line) and CR immunolabeled cells (green) in control (n = 7 zebrafish) and 3 dpi (n = 6 zebrafish). Arrowheads indicate all the NB+CR- cells. **b** The proportion of the NB+CR+ to NB+CR- cells in uninjured (control) and at 3 days post-injured animals. **c** The number of the NB+CR- neurons detected in the whole adult zebrafish spinal cord hemisegment in control and injured (3 dpi) animals. **d** Quantification of the NB+CR+ to NB+CR- soma sizes in uninjured and 3 days post-injured adult zebrafish. **e** Pseudocolored images and analysis of the normalized intensity of calretinin levels

in pMNs after application of saline or the non-selective gap-junction channel blocker carbenoxolone (CBX) obtained from uninjured (control) animals. **f** Images and analysis of pMN normalized CR expression following carbenoxolone in injured (3 dpi) zebrafish. Asterisks indicate the pMN cell bodies. μ population mean, CBX Carbenoxolone, CR calretinin, dpi days post-injury, NB neurobiotin (N-(2-ami-noethyl) biotinamide hydrochloride), SCI spinal cord injury, w/o without. Data are presented as mean ± s.e.m., as violin plots and as box plots showing the median with 25/75 percentile (box and line) and minimum–maximum (whiskers). *$P < 0.05$; **$P < 0.01$; ****$P < 0.0001$; ns, not significant. For detailed statistics, see Supple-mentary Table 1. Source data are provided as a Source Data file.

## Discussion

Here, we exposed the adaptive cellular responses to a traumatic insult in the adult zebrafish spinal cord. While our findings demonstrate that some of the cascades of cellular mechanisms and events initiated by injury are evolutionarily conserved between vertebrates, they have a profound differential outcome in adult zebrafish.

Our data reveal that in post-traumatic insult of the spinal cord, undamaged motoneurons proximal to the injury site receive an increased glutamatergic input that, in turn, raises their excitability and the cytosolic calcium (Ca²⁺) (Fig. 8). Although glutamate is constantly released into the synaptic cleft through both stimulated and sponta-neous exocytosis of synaptic vesicles, its concentration is tightly regulated by neurons and astrocytes[52,53]. Under pathophysiological conditions, such as injury, uncontrolled release of glutamate from damaged neurons, astrocytes, reactive microglia, and proteolysis into the parenchyma results in a significant increase in extracellular gluta-mate concentrations[4,5,10,51,54]. In our studies, we did not observe any tonic activation of the undamaged pMNs (as seen from the RMP; Fig. 3c and Supplementary Fig. 5) nor a change in the spontaneous synaptic activity after the application of TTX (Fig. 2f, g). Hence, our findings suggest that the excitatory drive may be caused by damaged or over-depolarized descending glutamatergic neurons within the injured area, but the rules governing this interaction remain unclear.

Although Ca²⁺ is an essential, versatile signal involved in a wide range of critical neuronal processes, such as gene expression, neuro-transmission, metabolism, and synaptic activity, sustained accumula-tion of unbuffered cytosolic Ca²⁺ could be harmful by triggering the cell death pathways[3,11,13,55]. In support, atypical expression of Ca²⁺-permeable AMPA receptors in mammalian spinal motoneurons caused neuronal degeneration[56]. Hence, the effective regulation of intracel-lular Ca²⁺ is a principal determinant of neuronal viability[29,40], and future studies are needed to determine the precise intracellular Ca²⁺ con-centration in zebrafish pMNs by ratio-metric approaches to define the upper limit that a neuron could process successfully. The less vulner-able or even resistant neurons can effectively regulate and buffer the intracellular Ca²⁺ through the use of calcium-buffering proteins such as Parvalbumin, calbindin-D28k, and Calretinin[22–26]. Accordingly, it is broadly accepted that the low presence of calcium-buffering proteins in mammalian spinal motoneurons[57–59] renders them vulnerable to

persistent excitability and calcium toxicity after injury or in neurode-generative conditions such as amyotrophic lateral sclerosis ALS[2,5,7,8,27–29]. Consistent with this notion, the extraocular (oculomotor nuclei) motoneurons that express abundant levels of Parvalbumin and calbindin D-28k show resilience in neurodegeneration[12,59]. Moreover, genetic overexpression of Parvalbumin in rodent motoneurons has been shown to protect them against pharmacologically or injury-induced excitotoxicity[59,60]. Our data here are consistent with such findings. While the fast pMNs express high levels of both Parvalbumin and Calretinin[41], our data here pinpoint the potentially crucial role of Calretinin during calcium dyshomeostasis. We not only showed the dynamic upregulation of the Calretinin expression in pMNs but also detected that 50% more neurons in the adult spinal cord start expressing Calretinin after injury. In support, a previous study showed the upregulation of the calretinin gene and proteins in spinal neurons that regenerate their axons in zebrafish[44]. Based on these findings, we conclude that Calretinin has an essential neuroprotective and restorative function in the adult zebrafish spinal cord.

We also revealed an adaptive mechanism by which spinal cord injury alters the connexin protein and gap junction channel home-ostasis (Fig. 8). Previous studies reported the post-injury induction of the proteins forming gap junction channels (connexins) in the nervous system[16–21]. However, the observed transient up-regulation of con-nexins reported here for the adult zebrafish spinal cord does not define their function. Thus, it was essential to parallel the connexin expression with the electrical and dye coupling experiments. Assessing the strength of electrical coupling (through the study of coupling coefficient) due to more gap junction channels is informative but may only sometimes be conclusive as additional mechanisms can affect gap junction channel permeability or their open state. However, when the data are consistent and mutually supportive, they provide valuable insights. Furthermore, gap junction proteins have diverse roles in neuronal function and tissue communication, acting as channels or hemichannels[61–63]. Gap junction channels are formed when two hemi-channels or connexons from adjacent cells dock together. Hemi-channels act as conduits in the cell membrane, enabling the transfer of molecules and ions between neurons and the extracellular space in a non-selective, diffuse manner. Our studies did not support the crucial role of neuronal hemichannels in the adaptive transformations of the

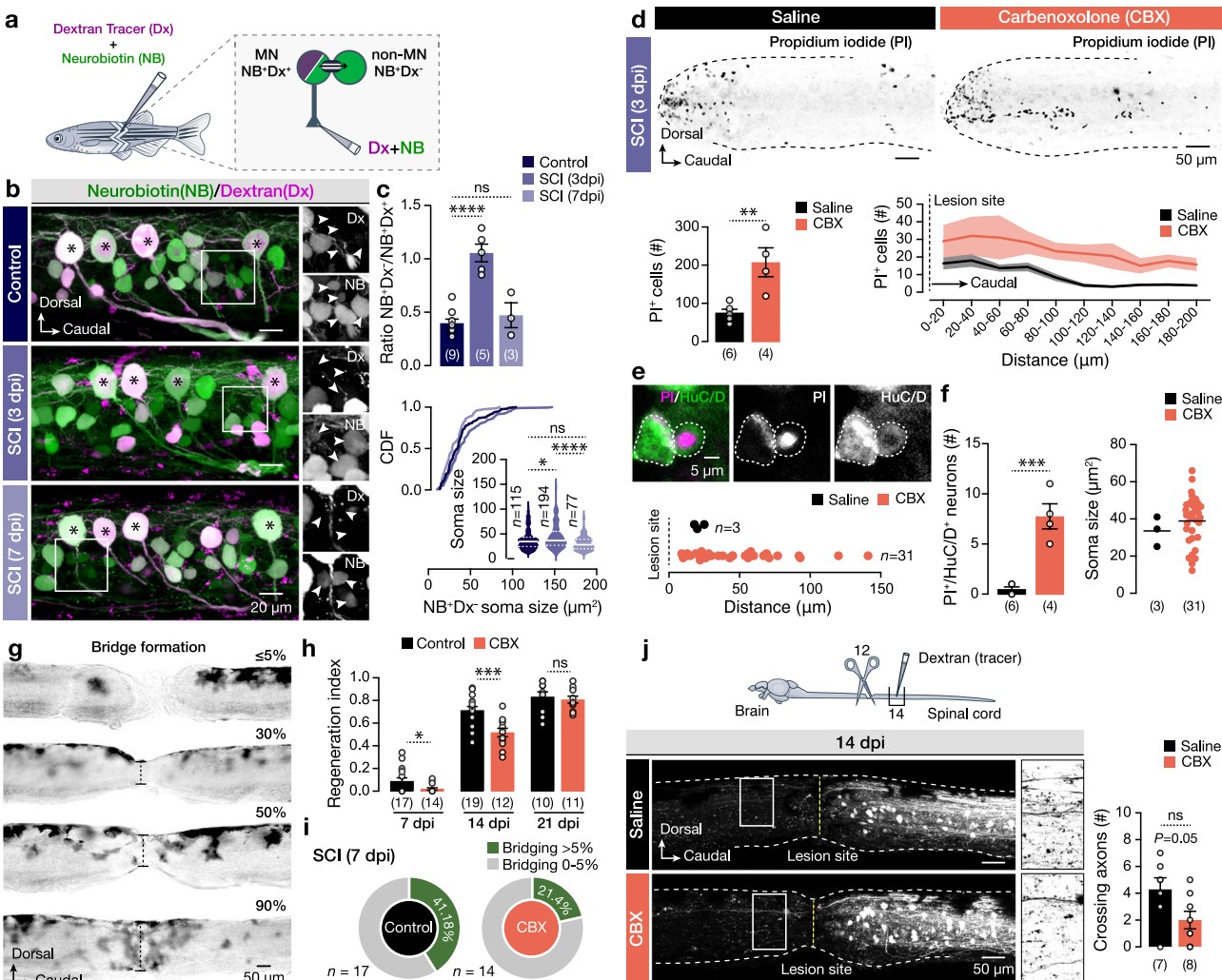

**Fig. 7 | Blockade of gap junction-mediated bystander changes affects neuronal death and initiation of the regeneration process. a** The experimental setup to investigate the bystander effect from the whole spinal MN population by injecting a mixture of NB (green) and Dextran tracer (Dx, magenta) into the ventral root and axial muscles. **b** Whole-mount confocal image (from a selected stack of images close to pMN level) of the adult zebrafish spinal cord showing the traced MNs (NB⁺Dx⁺) and the dye-coupled neurons (NB⁺Dx⁻). Arrowheads indicate the dye-coupled neurons (NB⁺Dx⁻). **c** Quantification of the ratio between dye-coupled neurons (NB⁺Dx⁻) to traced MNs (NB⁺Dx⁺) and the dye-coupled neurons (NB⁺Dx⁻) soma size in different experimental conditions. **d** Whole-mount inverted confocal images show PI⁺ cells (appear black) and their spatial distribution at 3 dpi after administration of the gap-junction channel blocker carbenoxolone (CBX). **e** Single-plane confocal images show the incorporation of PI (magenta) from spinal neurons (HuC/D⁺; green) and their relative distance from the lesion site. White dashed lines indicate the PI⁺HuC/D⁺ neurons. **f** Quantification and analysis of the number of PI⁺HuC/D⁺ neurons and soma sizes in saline and CBX at 3 dpi. **g** Progression (as %) of spinal cord bridge formation during the regeneration process. **h** Quantification of regeneration progress at different time points after injury in saline- (control) or CBX-treated animals. **i** At 7 dpi, carbenoxolone (CBX) delays the initiation of the bridge formation. **j** Experimental setup and images showing the neuronal processes that cross the lesion site at 14 dpi and analysis of the number of detected neurites. Asterisks indicate the pMN cell bodies. CBX Carbenoxolone, CDF cumulative distribution (frequency), Dx Dextran tracer, NB neurobiotin (N-(2-aminoethyl) biotinamide hydrochloride), PI propidium iodide, SCI spinal cord injury. Data are presented as mean ± s.e.m. and as box plots showing the median with 25/75 percentile (box and line) and minimum–maximum (whiskers). *$P < 0.05$; **$P < 0.01$; ***$P < 0.001$; ****$P < 0.0001$; ns, not significant. For detailed statistics, see Supplementary Table 1. Source data are provided as a Source Data file.

spinal cord network, similar to previous reports in rodents studying the bystander neuronal death after ischemia and NMDA-induced excitotoxicity[50]. In addition, gap junction channels exist in two main configurations: homotypic, involving identical hemichannels of connexins, and heterotypic, with different hemichannel assemblies[61]. The exact configuration is critical in determining molecule selectivity, unidirectional transfer, and rectification[61,62]. Whether the pMN gap junction channels are homotypic or heterotypic still remains unclear, yet current evidence suggests that their configuration in the zebrafish is neuron-to-neuron specific[47,63]. Collectively, our data are consistent with the model of the bystander effect, which proposes that adjacent cells become protected ("*Kiss of life*") or exposed ("*Kiss of death*") as a consequence of their proximity[10,14].

The outcome of the bystander effect in the adult zebrafish spinal cord after an injury does not promote cell death. We propose that the extended coupling between neurons assists in spreading cytotoxic molecules to neurons with effective buffering mechanisms such as the pMNs, thus reducing them to nontoxic levels (Fig. 8). Accordingly, gap junction channels not only allow the flow of electric current between neurons but also play a significant role in the intercellular exchange of a wide range of critical biochemical signals, such as inositol 1,4,5-trisphosphate (IP3), adenosine triphosphate (ATP), glucose, glutamate, cyclic adenosine monophosphate (cAMP), reactive oxygen species (ROS), small peptides, nucleotides and various ions including Ca²⁺[10,13–15,19,64,65]. However, our data cannot rule out the possibility that other potentially cytotoxic or neuroprotective molecules are also

involved in this exchange process between neurons. While we propose that diffusion of $Ca^{2+}$ contributes to neuronal survival, the $Ca^{2+}$ permeability through the gap junction channels is relatively low compared to calcium mobilizing second messengers such as the IP3 that has a significantly higher diffusion constant[64,65]. Thus, to fully understand the bystander effect in the spinal cord neurons, it will be necessary to investigate and reveal all the potential biochemical signaling molecules engaged in this process. However, this remains a significant technical challenge in investigations involving the whole tissue.

We note that regardless of the exact players mediating the bystander phenomenon, connexins and gap junction channels play an essential role in this process. Accordingly, the gap junction blocker carbenoxolone (CBX) induces cell and neuronal death in the post-traumatic zebrafish spinal cord. Consistent with this observation, previous studies showed that pharmacological blockade of the gap junction channels broadly protects neurons from the bystander "kiss of death" outcome[20,21,66].

Finally, the successful regeneration of the adult zebrafish spinal cord following traumatic insults is primarily attributed to its high level of neurogenic capacity[30–32]. Our findings here complement this view as we identified the precise mechanisms through which spinal cord circuits enact an important neuroprotective strategy after injury. We demonstrate that neurons particularly vulnerable in the mammalian spinal cord, such as the motoneurons[7,8,27–29,40], remain resistant to insults in zebrafish. Thus, our findings critically broaden our understanding of the diverse mechanisms (neurogenesis and neuroprotection) and their potential interplay that safeguards zebrafish's regeneration process.

## Methods

### Experimental animals

Adult (8–12 weeks old) wild-type zebrafish (*Danio rerio*; AB/Tübingen, RRID: ZIRC_ZL1) and Tg(*Chx10:GFP^nns1*) of both sexes were used. All zebrafish had similar sizes in each experimental procedure to minimize the variability due to size/age. No other selection criteria and blinding procedures were used to allocate animals to any experimental group. The Swedish Regional (Stockholm) ethical committee (Ethical permit no. 9248-2017; 19535-2020; 7650-2022) approved all experimental protocols and were implemented under the EU directive for the care and use of laboratory animals (2010/63/EU) and directed according to the ARRIVE guidelines. All efforts were made to use the minimum number of zebrafish needed to obtain reliable scientific data for statistical analyses.

### Spinal motoneuron tracing

Retrograde tracing of spinal motoneurons was performed in anesthetized animals with 0.03% tricaine methane sulfonate (MS-222; E10521, Sigma-Aldrich) by using dye injections with tetramethylrhodamine-dextran (3000 molecular weight; D3307, Thermo Fisher) or with biotinylated dextran (3000 molecular weight; D7135, Thermo Fisher) into the muscles or ventral roots of the corresponding myotomes. Injected zebrafish were kept for a minimum of 4 h to allow the retrograde transport of the tracer to the motoneuron soma. When the biotinylated dextran tracer was used, the spinal cords were thoroughly washed with PBS and fixed in 4% paraformaldehyde (PFA) in phosphate-buffered saline (PBS; 0.01 M, pH = 7.4; CAS30525-89-4, Santa Cruz Biotechnology, Inc.) overnight at 4 °C. Tissue was finally incubated with Alexa-conjugated streptavidin (dilution 1:500; *see* Supplementary Table 2) overnight at 4 °C, protected from light.

### Spinal cord injury

Animals were anesthetized in 0.03% tricaine methane sulfonate (MS-222; Sigma-Aldrich) before the complete transection of the spinal cord segment 12 with a micro knife (10318-14, Fine Science Tools) under continuous visual control through a stereo microscope (Leica Microsystem SD9, Leica). All transected animals were transferred and kept

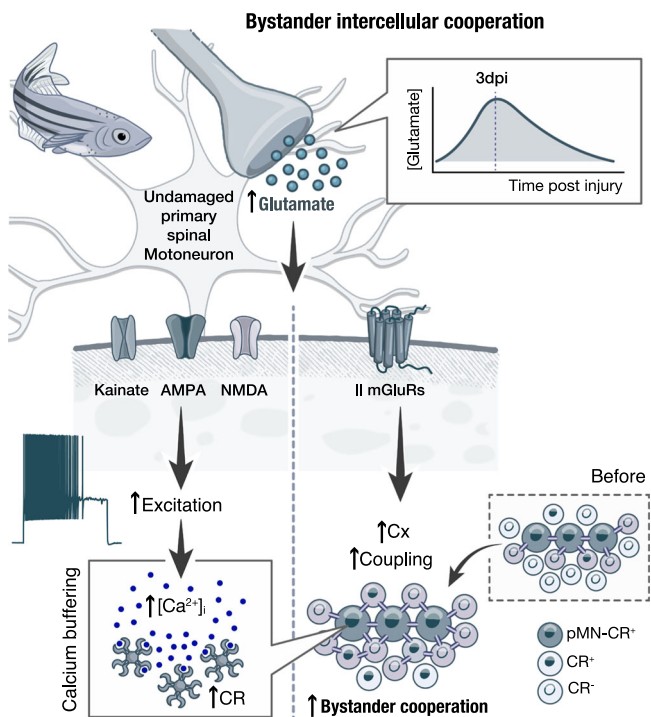

**Fig. 8 | Illustration of the proposed model for neuroprotective transformations in the adult zebrafish spinal cord after injury.** The results propose a complex and dynamic interplay among diverse cellular responses, including glutamatergic signaling, calcium buffering, and gap-junction-mediated bystander cooperation that ensure proper functioning and recovery of spinal cord neurons after injury.

under standard conditions in freshwater containing acetylsalicylic acid (Aspirin; 2,5 mg/Lit) for 24 h (1 dpi), 72 h (3 dpi), or 168 h (7 dpi).

### Propidium iodide labeling

To reveal the dying spinal cord cells after injury, animals were deeply anesthetized with 0.3-0.5% tricaine methane sulfonate (MS-222; E10521, Sigma-Aldrich) and dissected in ice-cold extracellular solution (see "Electrophysiological recordings" section). Axial muscles and vertebral arches were removed to expose and facilitate the permeation in the spinal cord tissue. Then, the spinal cord preparations were transferred in an extracellular solution containing 5% PI (Propidium Iodide; P4864, Sigma-Aldrich) and incubated at room temperature for 30 min with gentle agitation. The tissue was then rinsed three times for 10 min in fresh extracellular solution and fixed in 4% PFA overnight at 4 °C. After thorough rinses with PBS; tissues were mounted on glass slides in an 80% glycerol (G9012, Sigma-Aldrich) in PBS mounting solution. In some experiments (Supplementary Fig. 1), spinal MNs were first traced as described above (*see* "Spinal motoneuron tracing" section) and then processed for PI labeling.

### Morphological Reconstructions

Primary MNs (pMNs) were passively filled with 0.5-1% neurobiotin tracer (NB, N-(2-aminoethyl) biotinamide hydrochloride; SP-1120Vector Labs, SP-1120) in intracellular solution using the same methodological approach as that obtaining electrophysiological recordings (see "Electrophysiological recordings" section) to post-hoc expose and reconstruct their morphologies. After establishing a whole-cell configuration, the pMNs were kept in the recording chamber for 20–30 min to allow the complete diffusion of the tracer. The spinal cords were then removed, thoroughly washed with PBS, and fixed in 4% PFA overnight at 4 °C. Tissue was then incubated with Alexa-conjugated streptavidin (dilution 1:500, see supplementary table 2) overnight at 4 °C, sheltered from the light. After thoroughly

rinsing with PBS, tissue was mounted on glass slides in an 80% glycerol (G9012, Sigma-Aldrich) in PBS mounting solution.

## Immunohistochemistry

Immunolabeling was performed in both whole-mount spinal cords and cryosections. For cryosections, the tissue was cryoprotected overnight in 30% (w/v) sucrose in PBS at 4 °C, embedded in a cryosectioning medium (Cryomount; 45830, Histolab), rapidly frozen in dry-ice-cooled isopentane (2-methyl butane; 277258, Sigma-Aldrich) at −35 °C, and stored at −80 °C until use. The tissue's transverse coronal plane cryosections (thickness: 20–25 µm; Cryostat: Epredia CryoStar NX70, FisherScientific) were collected and processed for immunohistochemistry. In all whole-mount and cryosections immunohistochemistries, the tissue was washed 3 times for 5 min each in PBS. Nonspecific protein binding sites were blocked, and diluted primary antibodies (see Supplementary Table 2) were applied for 1–3 days at 4 °C, as previously described in detail[31,41,67,68]. After thorough rinses with PBS, the appropriate fluorescent secondary antibodies or biotinylated secondary antibodies were applied overnight at 4 °C. When a biotinylated secondary antibody was used, streptavidin conjugated to Alexa (see Supplementary Table 2) was applied overnight at 4 °C. Finally, the tissue was thoroughly rinsed and cover-slipped with a glycerol/PBS mounting solution.

## RNAscope in situ hybridization

RNAscope In situ hybridization experiments were performed according to the manufacturer's instructions in whole mound spinal cord zebrafish preparations. The mRNA for CR was detected using RNAscope (Advanced Cell Diagnostics) zebrafish-designed target probe *Calb2a*. Cell nuclei were revealed using the DAPI staining following the in situ hybridization.

## Electrophysiology

Zebrafish were deeply anesthetized in an ice-cold extracellular solution. The skin, axial muscles, and bones were removed to allow access to the spinal cord. The ex vivo spinal cord preparation was transferred to a recording chamber perfused with an extracellular solution containing 135.2 mM NaCl, 2.9 mM KCl, 2.1 mM CaCl2, 10 mM HEPES, and 10 mM glucose at pH 7.8. Single whole-cell intracellular recordings from pMNs were performed in voltage- and current-clamp modes. All electrodes (resistance: 5–7 MΩ) were pulled from borosilicate glass (outer diameter: 1.5 mm; inner diameter: 0.87 mm; Hilgenberg) on a micropipette puller (model P-97, Sutter Instruments) and filled with an intracellular solution containing 120 mM K-gluconate, 5 mM KCl, 10 mM HEPES, 4 mM Mg2ATP, 0.3 mM Na4GTP, and 10 mM Na-phosphocreatine at pH 7.4. Primary motoneurons (pMNs) and interneurons (V2a, GFP+) were visually identified using a fluorescent microscope (LNscope; Luigs & Neumann) equipped with a CCD camera (Lumenera) at 60x magnification. The electrodes were advanced to targeted pMNs or V2a-INs using a motorized micromanipulator (Luigs & Neumann) while applying constant positive pressure. Recorded signals were amplified with a MultiClamp 700B intracellular amplifier (Molecular Devices). During voltage-clamp recordings, all pMNs were clamped at −75 mV throughout the experiment. All experiments were performed at RT (~23 °C). For all dual whole-cell recordings of pMNs, two electrodes, as described before, were advanced from opposite directions into the spinal cord to record cells from the same spinal segment (intra-segmental recordings). The electrical properties were quantified in the absence of any bias current injection. The occurrence of electrical coupling was tested using hyperpolarizing current pulses (duration: 500 ms), and all dual whole-cell recordings are presented as averages of approximately 40 sweeps. Only pMNs that had stable resting membrane potentials at or below −50 mV and showed minimal changes in resistance (<5%) were included. In all recordings, the EPSC and EPSP postsynaptic events were detected and analyzed in a semi-automatic (supervised) fashion from the baseline using AxoGraph (version X 1.5.4; AxoGraph Scientific, Sydney, Australia; RRID: SCR_014284) or Clampfit (version 10.6; Molecular Devices, RRID: SCR_011323). The following drugs (prepared by diluting stock solutions in distilled water) were added (alone or in combinations as mentioned in the text) to the extracellular solution: TTX (tetrodotoxin, 1 µM; T8024, Sigma-Aldrich), NBQX (20 µM; N183, Sigma-Aldrich), and AP5 (50 µM; A5282, Sigma-Aldrich). The nonspecific blocker Carbenoxolone (200 µM; C4790, Sigma-Aldrich) was applied 45–60 min before the dual recordings to inhibit the gap-junction channels.

## Dye coupling

In dye coupling, undamaged pMNs (segment 14) were treated as described before for reconstructions of MN morphologies (see "Morphological Reconstructions" section). In this set of experiments, 0.2-0.5% NB solution was used to minimize the potential diffusion and absorption of the dye from damaged neuronal processes without affecting the number of labeled cells. To allow the complete diffuse of NB tracer through the gap junctions, the pMNs remain in the continuous perfusion of extracellular solution for at least 30 min at RT. To evaluate the effectiveness of Carbenoxolone (200 µM; C4790, Sigma-Aldrich) as a potent nonspecific gap-junction channel blocker, bath application of Carbenoxolone (200 µM; C4790, Sigma-Aldrich) applied for 30 min before injection of NB to pMN and the spinal cords remained to the chamber for another 45 min before processed. In all cases, spinal cords were dissected out, fixed, and streptavidin was applied as described above (*see* "Morphological Reconstructions" section). To evaluate the bystander effect from the whole MN population, two groups of animals were injured (as described above), and respectively left to survive for 3 and 7 days after the injury. 24 h before the sacrifice, injured and controls (uninjured) were injected into the muscles and ventral roots caudally to the lesion site (segment 12), with a mixture of NB (green) and Dextran tracer (Dx, magenta). After the sacrifice, the spinal cords were fixed in 4% PFA overnight at 4 °C and then carefully dissected. To reveal the NB labeling, the tissue was incubated with Alexa-conjugated streptavidin (dilution 1:500, see supplementary table 2) overnight at 4 °C, sheltered from the light. After thoroughly rinsing with PBS, the spinal cords were mounted on glass slides in an 80% glycerol (G9012, Sigma-Aldrich) in PBS mounting solution. Confocal images corresponding to the hemisegments 13 and 14 were taken.

## Pharmacological administrations

To test the contribution of glutamate receptors in the connexin expression (Fig. 3c), anesthetized uninjured adult zebrafish with 0.03% MS-222 and injected intraperitoneally with saline, a mixed solution of potent agonists for ionotropic glutamate receptors NMDA (50 µM; M3262, Sigma-Aldrich), and AMPA (30 µM; A9111, Sigma-Aldrich) or with the potent and selective agonist for the group II metabotropic glutamate receptors LY379268 ((-)-2-oxa-4-aminobicyclo[3.1.0]hexane-4,6-dicarboxylate, 12 µM; 2453, Tocris) for 24 h. After that, the animals were processed for immunodetection as described above (see Immunohistochemistry section). In the experiments shown in Fig. 3d, animals first received a spinal cord injury (see Spinal cord injury section) followed by three intraperitoneal injections of either the agonist LY379268 (12 µM; 2453, Tocris) or with the potent and selective orthosteric antagonist for the group II metabotropic glutamate receptors LY341495 ((1S,2S)-2-[(2S)-2-amino-3-(2,6-dioxo-3H-purin-9-yl)-1-hydroxy-1-oxopropan-2-yl]cyclopropane-1-carboxylic acid, 12 µM; 1209, Tocris) every 24 h and then processed for immunohistochemistry. To systemically block the gap junction channels, another group of uninjured experimental animals, after anesthesia, received an intraperitoneal injection of either saline or carbenoxolone solution (200 µM; C4790, Sigma-Aldrich) for 24 h and then processed for CR immunodetection. To evaluate the role of gap-junction channels in cell

death, transected adult zebrafish received three intraperitoneal administrations of carbenoxolone solution (200 µM; C4790, Sigma-Aldrich) every 24 h before being processed for a combination of PI detection with HuC/D immunohistochemistry as described above. To assess the potential cell toxicity of CBX administration into zebrafish spinal cord cells, uninjured animals were injected intraperitoneally with different dosages of CBX (200 µM, 1 mM, 5 mM, and 10 mM) and allowed to survive for 5 h, then processed for PI staining as described above (*see* "Propidium iodide labeling" section). In all cases described above, each intraperitoneal injection volume was always 2 µl.

### Tracing neuronal processes

To assess how blocking gap junctions affects early recovery, animals were injured and left to recover for 14 days while receiving intraperitoneal injections of CBX (200 µM; C4790, Sigma-Aldrich) every three days. 24 h before the sacrifice, a biotinylated dextran tracer (3000 molecular weight; D7135, Thermo Fisher) was injected into the spinal cord (segment 14) to reveal projections crossing the lesion site. The tracer was let to diffuse for 24 h until the sacrifice. Then, spinal cords were fixed and dissected. NB labeling was revealed after incubating the spinal cords with Alexa-conjugated streptavidin (dilution 1:500, see supplementary table 2) overnight at 4 °C, sheltered from the light. After thoroughly rinsing, spinal cords were mounted and imaged as described above (*see* "Immunohistochemistry" section).

### Calcium imaging

Retrograde loading of pMNs with a calcium indicator was performed using the Calcium Green-1 Dextran (3000 MW; C6765, ThermoFisher) after dissolving the tracer salt crystals in distilled water and injecting the generated paste into axial muscles. Multiple short-duration subthreshold current pulses were used to stimulate the spinal cord and to depolarize the pMNs via a glass pipette placed and attached to the dorsal side of the spinal cord. All analyzed pMNs were located 90-150 µm from the stimulation electrode. We used two different stimulation protocols: a train of 10 stimulation pulses (1 Hz, duration 10 sec) and a train of 15 pulses (30 Hz, duration 500 ms). Cells were visualized and recorded using a microscope (Zeiss Axioskop FS Plus, Zeiss) equipped with a CCD camera (Hamamatsu) at 80 or 100 ms/frame sampling rate. To determine the resting calcium concentration in pMNs, the cell-permeant calcium indicator dye Oregon Green 488 Bapta-1 AM (OGB-1 AM; O6807, ThermoFisher) was applied to spinal cords. The dye was first prepared as a stock solution (5 mM) by dissolving it with 20% w/v PF-127 (Pluronic F-127, ThermoFisher, P6867) in Dimethyl sulfoxide (DMSO Anhydrous, D12345, ThermoFisher). Adult zebrafish spinal cords were incubated with 12 µM OGB-1 solution for 24 h at RT, with gentle agitation. After thorough rinsing with PBS, the spinal cords were mounted and immediately imaged.

### Analysis

All immunodetections of whole-mount images of the adult zebrafish spinal cord preparations and cryosections were acquired using an LSM700 or LSM800 laser scanning confocal microscope (Zeiss) using either a dry 20x or an oil-immersed 40x objective. Each examined whole-mount spinal cord hemisegment was scanned from the ipsilateral side to the contralateral side after the central canal to ensure the presence of the whole hemisegment generating a z-stack (z-step size = 1–2 µm). Most of the spinal cord whole-mount images were generated using a subset of the original z-stacks close to pMN location to enhance the visualization of our presented data. Colocalizations were detected by visual identification of structures whose color reflects the combined contribution of two antibodies or markers in the merged image. The average fluorescence intensity (mean gray value) was measured in the manually drawn region of interest (ROI) in ImageJ (RRID: SCR_003070), around the neuronal soma from a single z-stack

frame that represents the epicenter of the neuron, usually with the largest diameter of the cell body. Differences in the background fluorescence level were corrected by subtracting the average intensity from an unstained image region. In some images, pseudo-coloring was performed by applying the fire LUT (ImageJ) to the corrected background images. Whole-mount preparations are used to calculate the relative position of spinal neurons through the spinal cord's lateral, dorsal, and ventral edges and the central canal as landmarks in ImageJ. The contour maps were constructed using OriginPro (v8, OriginLab; RRID: SCR_014212). For the Cx35/36 data, the presented images and all analyses of intensity and number of puncta were performed by merging a z-stack from the lateral side of neuronal soma to the epicenter, usually 4-6 image frames. After correcting the detection threshold, the number of aggregations (Cx35/36 puncta) was measured using the particle analysis tool in ImageJ. The assessed morphological features of neurons include the soma area (soma size, µm$^2$), dendritic branches (number), dendritic length (µm), and the extension of the area that dendrites cover in the spinal cord (µm$^2$). All morphological features and the number of cells were measured using ImageJ. Sholl analysis was performed using the Sholl Analysis Plugin for ImageJ (https://imagej.nih.gov/ij). The dye-coupled cells' spatial distribution and XY coordinates were obtained in ImageJ by correcting the distances after placing the filled pMN in the center. All final images were prepared with Adobe Photoshop (Adobe Systems Inc., San Jose, CA, USA; RRID: SCR_014199). To improve the visualization of our data, fluorescent images were inverted, presented as a single channel, or converted to magenta-green for color-blind readers. Digital modifications of the images (brightness and contrast) were minimal to diminish the potential distortion of biological information. Whole-mount spinal cord preparations often have pMNs in different focal planes, leading to misleading visualizations if presented as a whole hemisegment. Therefore Calretinin, Parvalbumin, *Calb2a*, Cx35/36 images are presented as a single pMN extracted from a distinct focal plane (midline, where the pMN soma has the largest diameter) or from a stack of images (most lateral focal plane of the soma to midline), as they used for analyses. All voltage clamp presented traces were low-pass filtered (Gaussian, 11-21 coefficients) using Clampfit (version 11.0; Molecular Devices). The EPSC amplitude was calculated as the difference between the baseline and the event's peak. All electrophysiology parameters were measured and analyzed using Clampfit (version 11.0; Molecular Devices). Phase plots represent the first action potential from a ramp depolarization protocol and are plotted as changes in membrane potential over time (dV/dt measured as mV/ms) against the instantaneous voltage value (mV). The generated loop graph was used to extract the action potential threshold value at which the dV/dt exceeded 12 mV/msec. The electrical cell-to-cell coupling strength here is expressed as the coupling coefficient (CC) in a steady state, which is the ratio of voltage deflection measured during the hyperpolarizing current injection from the recipient neuron to the stimulated neuron (Vm-recipient/Vm-stimulated). The adaptation index was calculated as an index of the firing duration during a depolarization step of 500 msec at 120% of the rheobase. Fluorometric quantification and analysis of the pMN calcium transients were performed after manually defining the soma size, generating a region of interest (ROI) in ImageJ. The relative fluorescence changes ($\Delta F/F_O$) after a background correction with baseline background fluorescence ($F_O$). In Oregon green 488 BAPTA-1 AM (OGB-1 AM) experiments, the average fluorescence intensity of each pMN of all pixels over its soma, including the nucleus, consistently had higher intensity than the cytosol. The intensity of OGB-1 AM was corrected to the background fluorescence of each experimental group (control, 3 dpi). The regeneration index (RI) was calculated as the proportion of the bridge formation coverage in relation to the dorsoventral axis of the adjacent uninjured spinal cord segments. For a bridging size of 5% or less, the RI was 0. All normalizations in the text and figures were performed for

each individual property to the highest value for that particular feature. All figures and graphs were prepared using Adobe Illustrator (Adobe Systems Inc., San Jose, CA, USA; RRID: SCR_010279), Prism (GraphPad Software Inc.; RRID: SCR_002798), and Microsoft Excel (spider plot; RRID: SCR_016137). To prevent observer, selection, and confirmation biases, experiments and analyses performed: (1) by another individual (analyst) than the one who collected the data (experimenter) using a pre-set unbiased computer-assisted image analysis system described above, (2) from two or more individuals in independent sets of experiments, (3) including a large set of animals in replicated experiments. In all cases, the individuals arrived at the same conclusions.

### Statistics and reproducibility
Parametric tests, such as the two-tailed (two-sided) unpaired or paired Student's *t*-test and *one-way* ANOVA (ordinary) followed by post hoc Tukey's test (for comparisons between all groups) or Dunnett's (for comparisons of groups to control group) multiple comparison tests, were used to identify the significance of differences between the experimental data group means. All statistics were performed using Prism (GraphPad Software Inc.). In all presented graphs, the significance level is indicated as $*P < 0.05$, $**P < 0.01$, $***P < 0.001$, and $****P < 0.0001$. n.s. indicates no significance. Analyzed data are presented as mean ± s.e.m. (standard error of the mean), as violin plots, as box plots showing the median, 25th, and 75th percentile (box and line) and minimal and maximal values (whiskers), or as spider plots. Finally, the *n* values indicate the final number of validated animals per group, cells, or events and are presented in detail in Supplementary Table 1. Most of the experiments were carried out independently 2–7 times by different investigators.

### Reporting summary
Further information on research design is available in the Nature Portfolio Reporting Summary linked to this article.

## Data availability
The data generated in this study are provided in the Source Data file. Detailed analyses of the data presented in this study are included in Supplementary Table 1. Source data are provided with this paper.

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

## Acknowledgements

The authors thank Dr. Andrea Nistri, Dr. George Mentis, Dr. Reinhard W. Köster, Dr. Marina Vidaki, and Ampatzis Lab members for their valuable discussion, comments, contributions to the project, and assistance in preparing this manuscript. This work was supported by grants from the Swedish Research Council (2015-03359 and 2020-00205 to K.A.), Anna-Stina och John Mattsons Minnesstiftelse för sonen Johan (FO2021-0041 to K.A.), StratNeuro (to K.A.), Swedish Brain Foundation (FO2019-0011 and FO2020-0003 to K.A.), Olle Engkvists Foundation (203-0003 to K.A.), Gösta Fraenckels Foundation (FS-2022:0006 and FS-2023:0005 to K.A.), Erik and Edith Fernström Foundation (FS-2019:0004 to K.A.) Karolinska Institute (to K.A. and WP.C.), and O.E. & Edla Johanssons Foundation (to WP.C.).

## Author contributions

K.A. conceived and designed the experiments. K.A. and WP.C. initiated the project. A.P., YW.ED., WP.C., L.D.V., and I.S. performed the electrophysiology experiments. A.P., L.L., and K.A. performed anatomy and pharmacology experiments. All authors collectively analyzed the data, discussed the results, and prepared the figures. K.A. wrote the manuscript with input from all authors.

## Funding

## Competing interests
The authors declare no competing interests.
