## [Peer Review File · Nature Communications]

Neuroprotective gap-junction-mediated bystander transformations in the adult zebrafish spinal cord after injuryREVIEWER COMMENTS

Reviewer #1 (Remarks to the Author):

In this manuscript by Pedroni et al., the authors propose a novel mechanism that may underlie the superior resistance of zebrafish motor neurons (MNs) to spinal cord injury (SCI). To support their hypothesis, the authors initially demonstrated that many primary MNs (pMNs) adjacent to the injury site, survived after injury, showed no cellular or morphological alterations. Subsequently, incorporating fine intracellular recordings, they revealed that excitatory postsynaptic potentials (EPSPs) of pMNs increased 3 days post-injury (dpi) and later returned to control levels at 7dpi. Interestingly, they also demonstrated that this increase in EPSPs post-injury is mediated by glutamatergic receptors but is resistant to TTX, indicating it is not due to neuronal firing of action potentials. The authors further quantified SCI-induced physiological changes in pMNs, showing that features of pMN excitability, such as input resistance, rheobase, and adaptation index, are largely modulated after SCI with an initial increase at 3dpi followed by a reduction to control levels at 7dpi. These results were supported by an increase in spontaneous and evoked calcium signals in pMNs. Additionally, the authors showed that the transient increase in neural excitability after SCI is accompanied by a transient increase in the calcium buffering protein, calretinin. This increase, they argue, protects pMNs from the damages of hyperexcitability after injury. Excitingly this SCI-induced boost in glutamate-driven excitability, calcium signals, and calretinin increase, accompanied by an increase in the expression of gap junction proteins Cx35/36 and functional coupling through gap junctions between motor neurons. Using pharmacology, histology, and physiology, the authors demonstrated that this increase in gap junction expression and coupling is specifically driven by the activation of metabotropic glutamate receptors mGluR2/3. Importantly, pharmacologically blocking gap junctions led to increased calretinin expression in MNs, but more significantly, it resulted in increased cell death and reduced regenerative capacity after SCI. Collectively, these results reveal that after SCI in zebrafish, increased calcium buffering and gap junction coupling can protect motor neurons from hyperexcitability-related cell death by spreading the impact of glutamate cytotoxicity through gap junction-coupled networks. Previously, the superb recovery of zebrafish from spinal cord injury has been attributed to their ability to generate new neurons. This manuscript proposes a novel and complementary mechanism, suggesting that zebrafish motor neurons are also more protected from SCI.

I am very excited about this study, it's one of the most innovative manuscripts I have reviewed in a long time. Congratulations to the authors for supporting their novel hypothesis with robust data, employing a range of methods from histology to pharmacology and carefully conducted physiological experiments. This combination is quite rare in the field of SCI, especially in the context of the zebrafish model. I also commend the authors for the well-written text and high-quality and clarity of their figures and data, making it easy to read through the manuscript and assess the importance of these results. As I read through the manuscript, I had many questions, and every time I moved to the next figure or results section, I found my questions answered with another carefully conducted experiment. This speaks very well to the completeness of this study, likely due to perhaps an earlier review in a different journal.

As mentioned earlier, I am very enthusiastic about this study and fully support its publication in Nature Communications. I have a few comments below that I hope will help the authors support their arguments and communicate better with the readers.

MAJOR COMMENTS

1. The authors successfully demonstrated through electrophysiological recordings that neuronal excitability and activity initially increase after SCI, later buffered by the increase in calretinin and gap junction. The calcium imaging experiments align with these results. However, measuring absolute levels of calcium with classical calcium imaging using Oregon Green BAPTA (OGB)-AM dye is challenging. It's hard to interpret the results in Figure 3g on OGB intensity, as different neurons can internalize different levels of the calcium indicator. To strengthen the claim about the role of increased calcium after SCI, the authors could consider using ratio metric calcium dyes such as Fura2-AM or Indo1-AM, allowing for the assessment of calcium levels, independent of labeling/internalization

intensity of calcium indicators. While the suitability of these ratiometric calcium dyes in zebrafish spinal cord is uncertain, attempting these experiments and reporting their success would enhance the study.

2. The presentation of calcium imaging results differs slightly from most studies using this method, with the authors choosing to normalize the calcium traces without a clear explanation. Providing more details on the normalization process would improve the assessment of these results. Additionally, showing and comparing the amplitude of $\Delta F/F$ after stimulation in Figure 3e/f, in addition to normalized comparisons, would increase confidence in these results.

3. Currently, only a representative example of spontaneous calcium traces is shown in Figure 3d. Would it not be important to quantify this spontaneous calcium fluctuations? Quantifying spontaneous calcium fluctuations, such as frequency, duration of bursts, or reporting the standard deviation of these calcium traces, could provide valuable quantification of this phenomenon. It is possible that none of this quantification will lead to significant alterations, as it is never easy to quantify spontaneous calcium, but I highly recommend the authors to take a look at papers recording spontaneous activity in zebrafish brain and see that can be done about analyzing this results. In fact given the authors exciting findings about increased gap junction coupling after SCI, one thing I would expect to see in spontaneous calcium fluctuations is to see an increase in correlations between calcium signals from MNs (or in general most spinal; cord neurons) after SCI. This can be done simply by calculating Pearson's correlations between neuronal calcium signals. Of course the durations of authors recordings might be too short to do such correlations based analysis, but if the authors have 1-2 minute long recordings this can further support authors conclusions on increased connectivity after SCI.

MINOR COMMENTS

- 1) Supp Fig3 is a very exciting result and should go on the main figure. Also I think that how TTX changes EPSP properties in control animals should also be presented in this main figure... I understand that the key point is to show that even in the presence of TTX there are still major AP5/NBQX sensitive EPSP. However, I am bit puzzled that we see many of the TTX insensitive EPSP's even in control animals. This reveals that a substantial input of pMN in control animals are still non action potential mediated. I guess this is also a novel finding. I think that the authors should at least discuss, where does these non AP dependent glutamatergic inputs come from in control animals, perhaps astroglia in spinal cord? perhaps in case of SCI the source is dying cells? Please add this to discussion.
- 2) how does the data in Figure 2d make a statement about a change or stability of glutamate receptor abundance? please explain better, or remove this statement.
- 3) what is the source of this SCI induced primary wave of glutamatergic drive. glia? dying neurons? please add to discussion.
- 4) the polar plot in Figure 3c is pretty but not easy to read or distinguish the colors. Please move those most important physiological measures (rheobase, Rinput, AP firing) and their comparisons from the supplemental figures to main main figure 3.
- 5) In general, many panels are referenced by a single panel reference (such as Figure 4g, also parts of Figure 5). however there are more information graphs on that single panel and sometimes it is hard to know where to look in the figure. Please consider splitting the panel names a bit more so that readers can be guided to the right graph in distinct panels. This applies to also few other figures too.
- 6) I did not understand what is the underlying cause of the observation of Fig 6d-e., how/why the addition of CBX lead to increase of CR? Can you explain or discuss this better please?
- 7) In discussion section, I would not argue against the notion that there is neuro-regeneration, but perhaps propose the findings of this manuscript as a novel mechanism complementing other processes supporting recovery after SCI.
- 8) Figure 3d, "Green-1" is probably Oregon-Green-Bapta? please correct this wording here and clarify what indicator in the figure legend.
- 9) There might be typo in the main text when citing Figure 3 e-f, in between parentheses. This should be e and f, not i and j. I think...
- 10) please split figure legends for Fig 6 b-c, and explain separately and in more detail.

Reviewer #2 (Remarks to the Author):

SUMMARY

The manuscript by Pedroni and colleagues explores the neuroprotective role of gap junctions following spinal cord injury in adult zebrafish. The work is motivated by our relatively poor understanding of the adaptive cellular mechanisms responsible for successful regeneration, beyond neurogenesis. The authors use a variety of anatomical and functional approaches, including dye fills, immunos, calcium imaging, electrophysiology, pharmacology and targeted lesions to address this issue in adult zebrafish, because of the well-documented regenerative capacity of their spinal cord. In mammalian spinal cord, following primary mechanical damage there is a secondary wave of neuronal degeneration that has the greatest impact on the largest motor units. The idea is that glutamate excitotoxicity via calcium homeostasis dysregulation triggers cell death pathways and the lack of calcium binding proteins makes them selectively vulnerable. There is also an upregulation of gap junctions, which allows cell death signals to propagate to neighboring motor neurons – something called a bystander effect. Through an elegant and complementary set of experiments, the authors demonstrate that in zebrafish the bystander effect is neuroprotective, rather than neurogenerative. Specifically, increased metabotropic glutamate receptor activation following local spinal damage triggers increased coupling that helps nearby fast units distribute relatively high levels of calretinin to motor neurons that lack it, buffering against glutamate excitotoxicity. The work provides a fascinating example of the same molecular pathways leading to the opposite outcome, neuroprotection vs. neurodegeneration, based on the presence or absence of the calcium-binding protein calretinin. The findings also provide a clear chain of events and convincing explanation for the different outcomes in zebrafish and mammals. The figures are beautiful and the manuscript is very well written. I only have a few comments for clarification that should bolster their case.

SPECIFIC COMMENTS

1. I'm trying to reconcile the observations that coupling increases, yet input resistance increases and rheobase decreases. From the current injection data, it looks like the gap junctions are non-rectifying. So, I was expecting that increases in gap junctions would decrease apparent input resistance by providing more paths for current to leak out. Do you have any potential explanation for this discrepancy? It's a little puzzling, but not deal breaking.
2. I'm also trying to reconcile the observations in Supplementary Fig. 6 and the real Fig. 6. CR expression increases in neighboring neurons in the supplement, but does not appear to in the in the real. Instead, the majority of NB positive neurons (thus coupled) are CR negative. I was expecting an increase in NB+/CR+, because it is spreading through gaps? Apologies if I'm missing something here, but I could use some clarification.
3. Figure 1a: Here and elsewhere, somata are labeled with asterisks, but the legend doesn't explain what they are meant to be indicating.
4. Figure 4c: SCI 3 dpf, top Vm/Im, the current trace is not properly lined to the voltage trace.
5. Figure 4d: What is the micro symbol representing in this panel and elsewhere? Note in the legend?
6. Figure 4e: The neurobiotin labeling for dye coupling illustrated here is not at all convincing. Circles obscure the somata, so it's difficult to make them out. Instead, labeling looks more filamental than somatic. Perhaps using arrowheads, so the outlines of dimly labeled somata are more obvious would improve things? How sure are you that these are not just processes are nicked on the way in? More

convincing examples need to be provided here to support the analysis.

7. Figure 7: Since dextran and neurobiotin are not the same molecular weight, how do you know that NB+/DX- cells are simply ones that smaller molecular weight neurobiotin filled more easily in your axial/ventral root injections? So nothing to do with coupling at all? Would it be better to focus on interneurons, which are unlikely to have been labeled by muscle/nerve injections? Something to moderate this concern would be helpful.

8. L436: This sentence seems unnecessarily provocative. To me, it seems like past work has revealed the contribution of neurogenesis to regeneration, while your work reveals a novel mechanism that helps promote survival of existing neurons. I wouldn't describe your findings as 'arguing against' anything, but rather extending them in a complementary way.

Reviewer #3 (Remarks to the Author):

Pedroni et al. have investigated cellular mechanisms that may support the remarkable ability of the adult zebrafish spinal cord to regenerate after complete spinal cord transection. They focused on the survival and sustained viability of large, rapidly-fatigable primary spinal motoneurons (pMNs) distal to the injury. As observed in studies of responses to spinal cord injury in other animals, they found that injury causes increases in extracellular glutamate that stimulate glutamatergic input to these neurons via AMPA and NMDA receptors, causes changes in their electrical excitability, and causes increase in their intracellular calcium following spinal cord stimulation. The increases in glutamatergic input and electrical excitability at 3dpi returned to control by 7dpi.

The level of parvalbumin remained constant in pMNs following spinal cord injury, but the level of calretinin increased by 1dpi and Calbindin mRNA (Calb2a) increased. Connexin Cx35/36 expression in pMNs and electrical coupling between pMNs also increased by 1dpi, peaked at 3dpi and declined by 7dpi. Injection of single pMNs with neurobiotin revealed dye coupling to increased numbers of other spinal neuron that peaked at 3dpi and returned to control numbers by 7dpi. As observed previously in rodents, the increases in connexin expression and electrical coupling depended on activation of mGluR2/3.

To test the bystander hypothesis that increased dye coupling of pMNs (CR+) to other neurons (CR-) buffers the calcium in the other neurons, the authors evaluated CR immunofluorescence intensity after treatment with a gap junction blocker carbenoxolone. CR intensity in pMNs was increased and the intensity of CR immunofluorescence was rescued in pMNs of injured fish. Neurons coupled to pMNs included v2a interneurons, suggesting that pMNs may rescue multiple populations of neurons. Treatment with carbenoxolone led to increased cell death of neurons with a range of sizes, assessed by PI incorporation and bridging of rostral and caudal parts of the spinal cord was delayed at 7 and 14 dpi.

Overall this is a thorough investigation. However there are a few points that should be clarified.

Major concerns

1. It is puzzling to observe increased coupling between pMNs and other spinal neurons when connexin 35/36 is expressed only in pMNs. Are other connexins expressed in sMNs and other neurons? Are heterotypic gap junctions formed? What connexins are partnering with Cx35/36? Connexins only in pMNs would argue for hemichannels, lines 267-277 notwithstanding. This point needs to be resolved

2. Does CR expression increase after injury in sMNs and other spinal neurons? Figure 3h,i and Supplementary Figure 6 suggest that it might be, but this is not explicitly stated.

3. The increase in $[Ca]_i$ in the other neurons (including V2a interneurons; these data were not included in Fig 3) needs to be quantified, since the test of the bystander hypothesis depends on elevation of $[Ca]_i$ in these other neurons.

4. The authors refer to "overuse" of the calretinin antibody (line 220) as the basis for a reduction in immunofluorescence of the antibody in Figure 3h. The conventional interpretation of immunofluorescence of an antibody is that it reflects the number of antigen binding sites and thus is proportional to the amount of the antigen (as in Supplementary Figure 6). Here it seems that the interpretation is not that the immunofluorescence refers to the amount of calretinin but to its ability to bind calcium. This would in principle depend on the binding site of the fluorophore and its proximity to the 3-D change in protein structure upon binding to calcium. Whether this Svant 6B3 antibody can report calcium independent of the amount of calretinin is not justified by citation or experiment in the manuscript. This point should be clarified. The continued presence of Calb2a mRNA is good to see, but does not in itself indicate the increased production of CR, since translation from mRNA to protein is a regulated step.

5. The authors place a good deal of trust in carbenoxolone as a specific blocker of electrical and dye coupling. Since specific blockade is central to the test of the bystander hypothesis, it would be helpful to know whether the major results are replicated with meclofenamic acid, a different allegedly specific blocker. It is gratifying that 200 μ M CBX does not reveal PI+ cells. Death is probably the most extreme form of toxicity. Two blockers, each with different side effects or mild forms of toxicity, would make a more compelling case for the specificity of blockade.

Other points

Figure 1c. Please label the home segment (14?).

Figure 2a. Please make the dashed lines white or change the figure legend.

Figure 3e,f. Where are I and j? Are they relics of previous lettering?

Figure 3g-i. What is μ about normalized intensity? Here and elsewhere, does μ signify "10⁻³"?

Lines 249,250. The coupling coefficient was not dependent on the distance between the neurons "over the distance tested (80 μ m)" or "within a spinal segment". The length constant is apparently long with respect to the length of a segment.

Figure 6a. Some white arrowheads do not seem to identify NB+CR- cell bodies.

Supplementary Figure 7a. Are the + and - assignments inverted in the center panel of the bottom row?

Supplementary Figure 10 title. It is not known whether CBX is without toxicity (adverse effects). It is shown that CBX affects pMN electrical and dye coupling without inducing PI uptake.

Lines 351,352. ..."demonstrate that some of the cascade"..."vertebrates, the novel features have a profound"...

Point-by-Point Response to Reviewers comments:

We are grateful for the positive comments and constructive criticism provided by the Reviewers. We have revised the manuscript, considering all comments and suggestions, as detailed below. The Reviewers' comments are indicated in bold and italics, while our responses are in regular style in this document. Moreover, all the amended text is highlighted in the revised manuscript file to allow the Reviewers to track our changes.

Reviewer #1:

In this manuscript by Pedroni et al., the authors propose a novel mechanism that may underlie the superior resistance of zebrafish motor neurons (MNs) to spinal cord injury (SCI). To support their hypothesis, the authors initially demonstrated that many primary MNs (pMNs) adjacent to the injury site, survived after injury, showed no cellular or morphological alterations. Subsequently, incorporating fine intracellular recordings, they revealed that excitatory postsynaptic potentials (EPSPs) of pMNs increased 3 days post-injury (dpi) and later returned to control levels at 7dpi. Interestingly, they also demonstrated that this increase in EPSPs post-injury is mediated by glutamatergic receptors but is resistant to TTX, indicating it is not due to neuronal firing of action potentials. The authors further quantified SCI-induced physiological changes in pMNs, showing that features of pMN excitability, such as input resistance, rheobase, and adaptation index, are largely modulated after SCI with an initial increase at 3dpi followed by a reduction to control levels at 7dpi. These results were supported by an increase in spontaneous and evoked calcium signals in pMNs. Additionally, the authors showed that the transient increase in neural excitability after SCI is accompanied by a transient increase in the calcium buffering protein, calretinin. This increase, they argue, protects pMNs from the damages of hyperexcitability after injury. Excitingly this SCI-induced boost in glutamate-driven excitability, calcium signals, and calretinin increase, accompanied by an increase in the expression of gap junction proteins Cx35/36 and functional coupling through gap junctions between motor neurons. Using pharmacology, histology, and physiology, the authors demonstrated that this increase in gap junction expression and coupling is specifically driven by the activation of metabotropic glutamate receptors mGluR2/3. Importantly, pharmacologically blocking gap junctions led to increased calretinin expression in MNs, but more significantly, it resulted in increased cell death and reduced regenerative capacity after SCI.

Collectively, these results reveal that after SCI in zebrafish, increased calcium buffering and gap junction coupling can protect motor neurons from hyperexcitability-related cell death by spreading the impact of glutamate cytotoxicity through gap junction-coupled networks. Previously, the superb recovery of zebrafish from spinal cord injury has been attributed to their ability to generate new neurons. This manuscript proposes a novel and complementary mechanism, suggesting that zebrafish motor neurons are also more protected from SCI.

I am very excited about this study, it's one of the most innovative manuscripts I have reviewed in a long time. Congratulations to the authors for supporting their novel hypothesis with robust data, employing a range of methods from histology to pharmacology and carefully conducted physiological experiments. This combination is quite rare in the field of SCI, especially in the context of the zebrafish model. I also commend the authors for the well-written text and high-quality and clarity of their figures and data, making it easy to read through the manuscript and assess the importance of these results. As I read through the manuscript, I had many questions, and every time I moved to the next figure or results section, I found my questions answered with another carefully conducted experiment. This speaks very well to the completeness of this study, likely due to perhaps an earlier review in a different journal.

As mentioned earlier, I am very enthusiastic about this study and fully support its publication in Nature Communications. I have a few comments below that I hope will help the authors support their arguments and communicate better with the readers.

Authors: We are pleased with the Reviewer's comments regarding the quality and importance of our work. We also acknowledge the Reviewer for considering our work suitable for publication.

MAJOR COMMENTS

1. The authors successfully demonstrated through electrophysiological recordings that neuronal excitability and activity initially increase after SCI, later buffered by the increase in calretinin and gap

junction. The calcium imaging experiments align with these results. However, measuring absolute levels of calcium with classical calcium imaging using Oregon Green BAPTA (OGB)-AM dye is challenging. It's hard to interpret the results in Figure 3g on OGB intensity, as different neurons can internalize different levels of the calcium indicator. To strengthen the claim about the role of increased calcium after SCI, the authors could consider using ratio metric calcium dyes such as Fura2-AM or Indo1-AM, allowing for the assessment of calcium levels, independent of labeling/internalization intensity of calcium indicators. While the suitability of these ratiometric calcium dyes in zebrafish spinal cord is uncertain, attempting these experiments and reporting their success would enhance the study.

Authors: We agree that the ratio-metric approach is the most appropriate method to verify the increased calcium (resting calcium) in the pMN population. As such, we attempted to address the comment raised by the Reviewer using the Fura-2 dye. Unfortunately, despite our best efforts, we were unable to obtain reliable delivery of the dye into the adult spinal pMNs. Initially, we tested the most appropriate method that complies with our research scope, which is the incubation of the tissue with the dye as we did before for the OGB-AM. The permeability of the dye through the adult spinal cord was unsuccessful, possibly due to the meninges' rigidity; even after prolonged incubation with chemicals that could increase the porosity of the tissue, we did not have reliable and specific staining. Moreover, we tried to deliver the dye by pressure micro-injections in the spinal cord as described before (Yaksi and Friedrich, 2006). While this approach is inappropriate for our studies, as the injection could be considered a form of injury of the spinal cord for the control animals, we yield minor and inconsistent staining of the spinal neurons that did not generate data. Our multiple unsuccessful attempts underscore the potential uncertainty surrounding the application of these dyes, which may also explain why they have not been used in the adult zebrafish spinal cord and limited use in zebrafish in general. However, we recognize the importance of accurately measuring the cytosolic Ca^{2+} concentration to determine a neuron's maximum processing capacity, and we acknowledge that our unsuccessful attempts have restricted the potential outcomes of our study. To this end, we have revised the discussion section of our manuscript to include this experimental limitation.

References:

Yaksi E, Friedrich RW. Reconstruction of firing rate changes across neuronal populations by temporally deconvolved Ca^{2+} imaging. Nat Methods. 2006 May;3(5):377-83. doi: 10.1038/nmeth874. PMID: 16628208.

2. The presentation of calcium imaging results differs slightly from most studies using this method, with the authors choosing to normalize the calcium traces without a clear explanation. Providing more details on the normalization process would improve the assessment of these results. Additionally, showing and comparing the amplitude of $\Delta F/F_0$ after stimulation in Figure 3e/f, in addition to normalized comparisons, would increase confidence in these results.

Authors: Here, all the normalizations of the data (for calcium imaging, immunohistochemistry intensity, etc.) were performed in relation to the highest value of the dataset (as we state in the "Material and Method" section). This is a normalization of the data vectors; as such, we rescale our dataset to a common scale or distribution of values without distorting differences in the range of data values, thus not changing the shape of the data. Our scope was to use as few as possible Arbitrary units. Hence, our calcium data normalization is independent of the **standardization** of the signals that aim to reveal the actual signal from the background. As such, the differences in $\Delta F/F_0$ values are not affected (see below, Figure 1-for Reviewers).

Figure 1 - for Reviewers. Revised panels from the calcium imaging analysis after removing the normalization of the data.

Understanding the Reviewer's comment here and to increase the accessibility of our work, we decide to follow the recommendation and present the calcium imaging data as in most published studies as a percentage of $\Delta F/F_0$. As the reviewer can now see, there is no difference in the actual shape of the data and the statistics presented in the revised *Figure 3*.

3. Currently, only a representative example of spontaneous calcium traces is shown in Figure 3d. Would it not be important to quantify this spontaneous calcium fluctuations? Quantifying spontaneous calcium fluctuations, such as frequency, duration of bursts, or reporting the standard deviation of these calcium traces, could provide valuable quantification of this phenomenon. It is possible that none of this quantification will lead to significant alterations, as it is never easy to quantify spontaneous calcium, but I highly recommend the authors to take a look at papers recording spontaneous activity in zebrafish brain and see that can be done about analyzing this results. In fact given the authors exciting findings about increased gap junction coupling after SCI, one thing I would expect to see in spontaneous calcium fluctuations is to see an increase in correlations between calcium signals from MNs (or in general most spinal; cord neurons) after SCI. This can be done simply by calculating Pearson's correlations between neuronal calcium signals. Of course the durations of authors recordings might be too short to do such correlations based analysis, but if the authors have 1-2 minute long recordings this can further support authors conclusions on increased connectivity after SCI.

Authors: We thank the Reviewer for pointing out this missing analysis. While we agree that analysis of the spontaneous activity is challenging, our recordings (of ~80 sec) allowed us to perform basic analysis of the calcium events' frequency and amplitude, similar to the approach we use for our electrophysiological recordings. These data are shown now in the revised version of *Figure 3*.

MINOR COMMENTS

1) Supp Fig3 is a very exciting result and should go on the main figure. Also I think that how TTX changes EPSP properties in control animals should also be presented in this main figure... I understand that the key point is to show that even in the presence of TTX there are still major AP5/NBQX sensitive EPSP. However, I am bit puzzled that we see many of the TTX insensitive EPSP's even in control animals. This reveals that a substantial input of pMN in control animals are still non action potential mediated. I guess this is also a novel finding. I think that the authors should at least discuss, where does these non AP dependent glutamatergic inputs come from in control animals, perhaps astroglia in spinal cord? perhaps in case of SCI the source is dying cells? Please add this to discussion.

Authors: We followed the Reviewer's suggestion, and now we present the TTX-related experiments by showing the corresponding traces and analyses in control and after-injury (3 dpi) experimental conditions in *Figure 2f,g*. Moreover, we revised *Figure 2h* to include the drug administration of glutamate receptor blockers (NBQX and AP5) in control animals. Regarding the source of the glutamatergic input, please see below our response to minor comment #3.

2) how does the data in Figure 2d make a statement about a change or stability of glutamate receptor abundance? please explain better, or remove this statement.

Authors: We are grateful for pointing out this potential issue, as it is likely that our presented data do not demonstrate clearly the relative contribution of the different ionotropic glutamate receptors in mediating the received input. Analysis of the proportion of the signals that are blocked after application of NBQX (for AMPA and Kainate receptors) and AP5 (for NMDA receptors) is similar for all primary MNs in all experimental conditions. Accordingly, in all cases, 60% of the signal is mediated through AMPA and Kainate receptors and ~28% through the NMDA receptors, regardless of the experimental condition control vs. SCI (*Figure 2 – for Reviewers*). This data demonstrates that we do not have changes in the relative abundance of the receptors after injury, as injury, for example, could change the abundance of NMDA but not of AMPA, which is not the case here. While this analysis is derived from the corresponding panels of *Figure 2i,j*, we understand that including this analysis in the manuscript will support our statements. Therefore, we now include this analysis as a new *Supplementary Figure 4*.

Figure 2- for Reviewers. Analysis of the proportion of the pMN glutamate receptor contribution in blockage of the postsynaptic events (EPSCs).

3) what is the source of this SCI induced primary wave of glutamatergic drive. glia ? dying neurons ? please add to discussion.

Authors: This is a fascinating question that we are currently working on to reveal the sources of glutamate in the injured spinal cord. While we do not have the precise answer to this question, there are several candidates that could provide the glutamate, as we illustrate below (see *Figure 3 – for Reviewers*).

Dying or damaged (axotomized after injury), glutamatergic neurons can release the produced glutamate. Moreover, GABAergic neurons also synthesize glutamate, which is a precursor of GABA. Glial cells such as astroglia store and have the ability to release glutamate upon cell injury (Parpura et al., 1994). One of the hallmarks of secondary injury is inflammation (Kuzhandaivel et al., 2011; Sámano and Nistri, 2019), which activates microglia that release massive amounts of glutamate (Barger et al., 2007; Domercq et al., 2013). Besides the apparent involvement of neurons and glia, glutamate also exists in the cerebrospinal fluid (CSF). In the case of our interventions, the complete spinal cord injury and disruption of the central canal allow the diffusion of the CSF content to the surrounding tissue. Moreover, in pathological conditions, it has been reported that the CSF contains high levels of glutamate (Madeira et al., 2018; Demartini et al., 2020). Finally, glutamate is an amino acid that is present in most of the proteins in all tissues besides the neuronal ones. As such, any protein deconstruction to amino acids (Proteolysis) will also provide additional free glutamate (Brosnan and Brosnan, 2013).

Figure 3 - for Reviewers. Schematic presentation of the potential sources of glutamate after injury.

However, in the context of our study, we think that these sources will heavily affect the neurons proximal to the injury site and less the neurons a bit distal to the injured area, such as the uninjured pMNs (studied here). A confirmation of that is that the pMN resting membrane potential does not change after injury. Based on this, the model that most probably describes our studies is that proximal to the injured area are many glutamatergic descending neurons that, along with the other neurons, will have non-contextual depolarization that will result in non-action potential mediated glutamate release (as we see from our TTX studies) (see also *Figure 3 – for Reviewers*). Following the Reviewer's comment, we have revised our text to accommodate some of the above information in our discussion section.

References:

- Barger SW, Goodwin ME, Porter MM, Beggs ML. Glutamate release from activated microglia requires the oxidative burst and lipid peroxidation. *J Neurochem.* 2007 Jun;101(5):1205-13. doi: 10.1111/j.1471-4159.2007.04487.x. Epub 2007 Mar 30. PMID: 17403030; PMCID: PMC1949347.
- Brosnan JT, Brosnan ME. Glutamate: a truly functional amino acid. *Amino Acids.* 2013 Sep;45(3):413-8. doi: 10.1007/s00726-012-1280-4. Epub 2012 Apr 18. PMID: 22526238.
- Demartini B, Invernizzi RW, Campiglio L, Bocci T, D'Arrigo A, Arighi A, Sciacca F, Galimberti D, Scarpini E, Gambini O, Priori A. Cerebrospinal fluid glutamate changes in functional movement disorders. *NPJ Parkinsons Dis.* 2020 Dec 4;6(1):37. doi: 10.1038/s41531-020-00140-z. PMID: 33298941; PMCID: PMC7718900.
- Domercq M, Vázquez-Villoldo N, Matute C. Neurotransmitter signaling in the pathophysiology of microglia. *Front Cell Neurosci.* 2013 Apr 19;7:49. doi: 10.3389/fncel.2013.00049. Erratum in: *Front Cell Neurosci.* 2013;7:107. PMID: 23626522; PMCID: PMC3630369.
- Kuzhandaivel A, Nistri A, Mazzone GL, Mladinic M. Molecular Mechanisms Underlying Cell Death in Spinal Networks in Relation to Locomotor Activity After Acute Injury in vitro. *Front Cell Neurosci.* 2011 Jun 17;5:9. doi: 10.3389/fncel.2011.00009. PMID: 21734866; PMCID: PMC3119860.
- Madeira C, Vargas-Lopes C, Brandão CO, Reis T, Laks J, Panizzutti R, Ferreira ST. Elevated Glutamate and Glutamine Levels in the Cerebrospinal Fluid of Patients With Probable Alzheimer's Disease and Depression. *Front Psychiatry.* 2018 Nov 6;9:561. doi: 10.3389/fpsy.2018.00561. PMID: 30459657; PMCID: PMC6232456.
- Parpura V, Basarsky TA, Liu F, Jęftinija K, Jęftinija S, Haydon PG. Glutamate-mediated astrocyte-neuron signalling. *Nature.* 1994 Jun 30;369(6483):744-7. doi: 10.1038/369744a0. PMID: 7911978.
- Sámano C, Nistri A. Mechanism of Neuroprotection Against Experimental Spinal Cord Injury by Riluzole or Methylprednisolone. *Neurochem Res.* 2019 Jan;44(1):200-213. doi: 10.1007/s11064-017-2459-6. Epub 2017 Dec 30. PMID: 29290040.

4) the polar plot in Figure 3c is pretty but not easy to read or distinguish the colors. Please move those most important physiological measures (rheobase, Rinput, AP firing) and their comparisons from the supplemental figures to main main figure 3.

Authors: Our spider plot aimed to provide a general overview of the physiological changes of the pMNs principal properties to a broad audience without electrophysiological knowledge, yet we see the point raised by the Reviewer. As such, we revised *Figure 3* accordingly; the spider plot is moved to supplementary material (new Supplementary Figure 5), and the panel is replaced with box plots for RMP (resting membrane potential), Rinput, and Rheobase. We consider the RMP important as it is the factor that determines the importance of Rinput and Rheobase changes; in other words, if we detected changes in the RMP between the experimental groups, then there will be other explanations for the differences in the alteration of the Rinput and Rheobase.

5) In general, many panels are referenced by a single panel reference (such as Figure 4g, also parts of Figure 5). however there are more information graphs on that single panel and sometimes it is hard to know where to look in the figure. Please consider splitting the panel names a bit more so that readers can be guided to the right graph in distinct panels. This applies to also few other figures too.

Authors: We originally expected that fewer panel divisions would simplify the extensive experiments presented here. Now, we revised all the Figures and divided the panels as much as necessary to accommodate more information in the Figure legends.

6) I did not understand what is the underlying cause of the observation of Fig 6d-e., how/why the addition of CBX lead to increase of CR ? Can you explain or discuss this better please ?

Authors: Indeed, this was possibly the most challenging task to describe sufficiently well in the text. Here, we found that the calcium increase in pMNs was buffered from CR (a protein that binds calcium quickly; Barinka and Druga, 2010), which increased in pMNs after injury. However, three days post-injury, we did not see this increase using antibody staining, although we observed an increase in the mRNA expression for CR. This suggests that our antibody detects predominantly the free CR (for more details, please see below the response to Reviewer#3, major concern #4). Our experiments led us to consider that calcium enters the pMNs through

gap-junction channels. We thought that blocking these channels with CBX would stop the flow of calcium to pMNs, increasing the CR expression because while the protein production is high, it is not used, and thus, our antibody could detect it. Our experiments confirmed this speculation that the antibody recognized predominantly the free CR, suggesting that calcium enters the pMNs through gap junction channels. If this were not the case, then the intensity of CR would remain the same before and after the blockage of gap junctions. We understand this data set is complicated, so we revised the manuscript text further to address this issue.

References:

Barinka F, Druga R. Calretinin expression in the mammalian neocortex: a review. *Physiol Res.* 2010;59(5):665-677. doi: 10.33549/physiolres.931930. Epub 2010 Apr 20. PMID: 20406030.

7) In discussion section, I would not argue against the notion that there is neuro-regeneration, but perhaps propose the findings of this manuscript as a novel mechanism complementing other processes supporting recovery after SCI.

Authors: Indeed, neuroprotection and neurogenesis are equally important and complementary mechanisms for the successful regeneration of the spinal cord, and thus, both could be very informative in translational studies for the potential regeneration and restoration of the mammalian spinal cord. While this statement sounds harsh, it is based on the prevailing view in the field of zebrafish spinal cord regeneration that zebrafish can successfully regenerate because of the amazing innate ability to generate new neurons, neglecting the involvement of neuroprotective strategies (as here and before this study) in the restoration process. Therefore, in our statement, we wanted to argue with this notion and suggest that both strategies are equally important. A similar comment was raised by Reviewer#2 (see Specific comments #8). Therefore, we decided to soften our statements in the revised version of our manuscript.

8) Figure 3d, “Green-1” is probably Oregon-Green-Bapta ? please correct this wording here and clarify what indicator in the figure legend.

Authors: In all experiments performed in *Figure 3* regarding the spontaneous calcium activity and calcium changes after stimulation, we used, as specified, the calcium indicator, Calcium Green-1 Dextran, injected into the axial musculature to achieve scholastic but specific retrograde labeling and evaluation of the spinal motoneurons (MNs), including the pMNs (indicated in the corresponding image with a star). This method allowed us to study the calcium changes in the MNs exclusively. Calcium Green-1 Dextran, like all the other indicators, exhibits an increase in fluorescence upon binding to Ca^{2+} and can allow up to a 14-fold increase in fluorescence intensity upon binding to Ca^{2+} . However, this indicator cannot provide a reading of the resting signal; thus, we used the Oregon Green BAPTA-1 AM for the resting calcium.

9) There might be typo in the main text when citing Figure 3 e-f, in between parentheses. This should be e and f, not i and j. I think...

Authors: Thank you. It has been corrected now.

10) please split figure legends for Fig 6 b-c, and explain separately and in more detail.

Authors: Done. Also, as the reviewer suggested, we divided most figure panels and legends to include more information.

Reviewer #2:

SUMMARY

The manuscript by Pedroni and colleagues explores the neuroprotective role of gap junctions following spinal cord injury in adult zebrafish. The work is motivated by our relatively poor understanding of the adaptive cellular mechanisms responsible for successful regeneration, beyond neurogenesis. The authors use a variety of anatomical and functional approaches, including dye fills, immunos, calcium imaging, electrophysiology, pharmacology and targeted lesions to address this issue in adult zebrafish, because of the well-documented regenerative capacity of their spinal cord. In mammalian spinal cord, following primary mechanical damage there is a secondary wave of neuronal degeneration that has the greatest impact on the largest motor units. The idea is that glutamate excitotoxicity via calcium homeostasis dysregulation triggers cell death pathways and the lack of calcium binding proteins makes them selectively vulnerable. There is also an upregulation of gap junctions, which allows cell death

signals to propagate to neighboring motor neurons – something called a bystander effect. Through an elegant and complementary set of experiments, the authors demonstrate that in zebrafish the bystander effect is neuroprotective, rather than neurogenerative. Specifically, increased metabotropic glutamate receptor activation following local spinal damage triggers increased coupling that helps nearby fast units distribute relatively high levels of calretinin to motor neurons that lack it, buffering against glutamate excitotoxicity. The work provides a fascinating example of the same molecular pathways leading to the opposite outcome, neuroprotection vs. neurodegeneration, based on the presence or absence of the calcium-binding protein calretinin. The findings also provide a clear chain of events and convincing explanation for the different outcomes in zebrafish and mammals. The figures are beautiful and the manuscript is very well written. I only have a few comments for clarification that should bolster their case.

Authors:

We are very thankful for the Reviewer's positive criticism regarding the quality and significance of our work. We also acknowledge the helpful comments that aim to increase the readability and accessibility of our studies.

SPECIFIC COMMENTS

1. I'm trying to reconcile the observations that coupling increases, yet input resistance increases and rheobase decreases. From the current injection data, it looks like the gap junctions are non-rectifying. So, I was expecting that increases in gap junctions would decrease apparent input resistance by providing more paths for current to leak out. Do you have any potential explanation for this discrepancy? It's a little puzzling, but not deal breaking.

Authors: This is a very interesting question raised by the Reviewer. Gap junctions provide low resistance (known as junctional resistance; R_j) pathways for ionic currents, facilitating fast and continuous communication between neurons. Similar to other membrane channels, they present voltage-gating processes, as their conductance is sensitive to the voltage difference between cells, known as trans-junctional voltage (V_j) (Curti et al., 2022). Gap junction channels exist in two main configurations: homotypic, involving identical hemichannels, and heterotypic, with different hemichannel assemblies. The homotypic gap junctions display a symmetrical dependency of conductance on V_j , whereas the heterotypic ones often but not always display asymmetrical relationships that yield diode-like electrical transmission (**rectification**) (Auerbach and Bennett, 1969; Phelan et al., 2008; Curti et al., 2022). Hence, to answer the Reviewer's question, we need to have a very clear understanding of how they are configured. Whether the pMN gap junction channels are homotypic or heterotypic still remains unclear as they are more likely neuron-type dependent. Similar to our response to Reviewer#3 regarding this topic (*please see* Response to Reviewer#3, major concerns #1) based on a recent study using zebrafish-specific antibodies against connexins (Miller et al., 2017), the authors found that only the Cx35.5 contributes to the gap junctions between V2a-INs and pMNs; thus they suggested a possible homotypic configuration (Pallucchi et al., 2022). Based on this, it is expected that the gap junction channels are non-rectifying. Still, as this is more likely a neuron-type specific property, more studies are needed in order to have a clear view and be able to address this question precisely.

It is also worth mentioning that the pMNs are the neurons with the lowest input resistance compared to any other neuronal type in the zebrafish spinal cord (McLean et al., 2007; Gabriel et al., 2011; Menelaou and McLean, 2012; Wang and McLean, 2014). As such, with increased interconnectivity after injury with many neurons of higher resistance, we expect a potentially increased flow toward the pMNs. Accordingly, as we observed in our experiments and presented in *Figure 4m* (revised manuscript), we found a higher coupling coefficient to the pMNs after injury. However, between pMNs, we did not detect any differences (*Supplementary Figure 10*), suggesting that the junctional electrical properties are neuron-to-neuron specific.

At this point, neither explanation is direct. Further studies are needed to clarify the configuration of the channels and the exact mechanisms utilized for the interconnectivity of neurons after injury.

References:

- Auerbach AA, Bennett MV. A rectifying electrotonic synapse in the central nervous system of a vertebrate. *J Gen Physiol.* 1969 Feb;53(2):211-37. doi: 10.1085/jgp.53.2.211. PMID: 4303657; PMCID: PMC2202903.
- Curti S, Davoine F, Dapino A. Function and Plasticity of Electrical Synapses in the Mammalian Brain: Role of Non-Junctional Mechanisms. *Biology (Basel).* 2022 Jan 5;11(1):81. doi: 10.3390/biology11010081. PMID: 35053079; PMCID: PMC8773336.
- Gabriel JP, Ausborn J, Ampatzis K, Mahmood R, Eklöf-Ljunggren E, El Manira A. Principles governing recruitment of motoneurons during swimming in zebrafish. *Nat Neurosci.* 2011 Jan;14(1):93-9. doi: 10.1038/nn.2704. Epub 2010 Nov 28. PMID: 21113162.

- McLean DL, Fan J, Higashijima S, Hale ME, Fetcho JR. A topographic map of recruitment in spinal cord. *Nature*. 2007 Mar 1;446(7131):71-5. doi: 10.1038/nature05588. PMID: 17330042.
- Menelaou E, McLean DL. A gradient in endogenous rhythmicity and oscillatory drive matches recruitment order in an axial motor pool. *J Neurosci*. 2012 Aug 8;32(32):10925-39. doi: 10.1523/JNEUROSCI.1809-12.2012. PMID: 22875927; PMCID: PMC3428065.
- Miller AC, Whitebitch AC, Shah AN, Marsden KC, Granato M, O'Brien J, Moens CB. A genetic basis for molecular asymmetry at vertebrate electrical synapses. *Elife*. 2017 May 22;6:e25364. doi: 10.7554/eLife.25364. PMID: 28530549; PMCID: PMC5462537.
- Pallucchi I, Bertuzzi M, Michel JC, Miller AC, El Manira A. Transformation of an early-established motor circuit during maturation in zebrafish. *Cell Rep*. 2022 Apr 12;39(2):110654. doi: 10.1016/j.celrep.2022.110654. PMID: 35417694; PMCID: PMC9071512.
- Phelan P, Goulding LA, Tam JL, Allen MJ, Dawber RJ, Davies JA, Bacon JP. Molecular mechanism of rectification at identified electrical synapses in the *Drosophila* giant fiber system. *Curr Biol*. 2008 Dec 23;18(24):1955-60. doi: 10.1016/j.cub.2008.10.067. Epub 2008 Dec 11. PMID: 19084406; PMCID: PMC2663713.
- Wang WC, McLean DL. Selective responses to tonic descending commands by temporal summation in a spinal motor pool. *Neuron*. 2014 Aug 6;83(3):708-21. doi: 10.1016/j.neuron.2014.06.021. Epub 2014 Jul 24. PMID: 25066087; PMCID: PMC4126198.

2. I'm also trying to reconcile the observations in Supplementary Fig. 6 and the real Fig. 6. CR expression increases in neighboring neurons in the supplement, but does not appear to in the in the real. Instead, the majority of NB positive neurons (thus coupled) are CR negative. I was expecting an increase in NB+/CR+, because it is spreading through gaps? Apologies if I'm missing something here, but I could use some clarification.

Authors: We apologize for this. Regardless of our efforts to present a complicated set of experiments and sequences of events simply, we were not always successful. Here, we found that the calcium increase in pMNs was buffered by the CR (fast calcium buffering). Moreover, our experiments point out the crucial role of the CR in buffering the increased calcium within the spinal cord in general. Collectively, our data suggest a model in which the spinal cord employs two different systems to cope with the increased calcium. More specifically:

- (1) The spinal neurons that before did not produce calretinin now start expressing calretinin to buffer the increased calcium. As a confirmation, we observed 50% more spinal neurons expressing CR after injury including some more fast secondary MNs (*Supplementary Figure 8*).
- (2) Neurons that are not capable of expressing CR, and thus less able to buffer the calcium, are now interconnected with neurons that have CR in order to send the calcium to these neurons for buffering. Among the neurons that have the ability to buffer excessive calcium, we found the pMNs. Accordingly, these neurons (pMNs), after injury, become interconnected mainly with neurons that do not have CR to help and support the buffering process.

Our data are important because they suggest a transient model (lasting for a few days) of an intraneuronal "community" that supports its members against injury. If we take a leap, we can view this as neuronal societal support for the weak and vulnerable members.

3. Figure 1a: Here and elsewhere, somata are labeled with asterisks, but the legend doesn't explain what they are meant to be indicating.

Authors: We apologize for our marking not being clear enough. As we stated in all figure legends, "Asterisks indicate the pMN cell bodies." We stated this for every figure legend before the description of the abbreviations and after the description of the panels, as this is a general marking reflecting collectively all the figures and not a particular panel.

4. Figure 4c: SCI 3 dpf, top Vm/Im, the current trace is not properly lined to the voltage trace.

Authors: Thank you. Now, it is corrected.

5. Figure 4d: What is the micro symbol representing in this panel and elsewhere? Note in the legend?

Authors: We apologize that we did not include the meaning of the μ meaning in the figure legends. The μ symbolizes the population mean of pMNs belonging to the same spinal cord and, thus, to the same animal.

6. Figure 4e: The neurobiotin labeling for dye coupling illustrated here is not at all convincing. Circles obscure the somata, so it's difficult to make them out. Instead, labeling looks more filamental than somatic. Perhaps using arrowheads, so the outlines of dimly labeled somata are more obvious would

improve things? How sure are you that these are not just processes are nicked on the way in? More convincing examples need to be provided here to support the analysis.

Authors: We understand the point raised by the Reviewer for the dye coupling images presented in our manuscript. The presentation of these data is challenging as all images derive from whole-mount preparations of the adult zebrafish spinal cord, in which we have several neurons with low NB transfer (dimly labeled somata). Replacing the images with others of similar resolution and labeling did not resolve the issue; as such, we decided to provide non-inverted, magenta fluorescent images in which the labeled neurons are marked by arrowheads as suggested by the Reviewer (see below *Figure 4 – for Reviewers*). Moreover, in most of our experiments, the tissue was processed for an immunodetection of the pan-neuronal marker HuC/D. Using this approach, we were able to identify the neuronal nature of the NB⁺ cells (*Figure 4 – for Reviewers*; Right panel).

Moreover, to avoid non-specific labeling, because of the potential damage of axons and dendrites of spinal neurons, we used two modifications. Firstly, we selected approaching the targeted pMN (for NB filling) using the minimum distance from the lateral surface of the spinal cord. Secondly, we paved the way with a pipette (electrode) that did not contain NB. After, we inserted a new electrode containing NB into the intracellular solution. To ensure this approach is correct, we performed some initial tests after inserting the NB-filled electrode without patching any neuron, and afterward, we processed these preparations with streptavidin labeling. In these experiments, we observed some labeled structures (neurites), yet we did not detect any NB⁺ neurons (HuC/D⁺).

Figure 4 - for Reviewers. (Left) the new presentation of dye coupling data as suggested by the Reviewer. (Right) NB spread after loading a single pMN (dashed white line). All NB⁺ cells are neurons (HuC/D⁺)

7. Figure 7: Since dextran and neurobiotin are not the same molecular weight, how do you know that NB+/DX- cells are simply ones that smaller molecular weight neurobiotin filled more easily in your axial/ventral root injections? So nothing to do with coupling at all? Would it be better to focus on interneurons, which are unlikely to have been labeled by muscle/nerve injections? Something to moderate this concern would be helpful.

Authors:

Indeed, this is a valid concern regarding our experimental approach. While we have an extensive experience in retrograde tracing the spinal cord MNs in adult zebrafish, we still cannot exclude the possibility that NB⁺DX⁻ MNs can be labeled using both of the tracers simultaneously, and that was initially one of our concerns. As such, we performed several experiments to verify that, in all cases, the number of labeled MNs with Dextran (only) or Dextran and NB (cocktail) was similar, given that there is animal-to-animal small variability of the number of motoneurons (see *Figure 5 – for Reviewers*). We also performed experiments in transgenic lines to verify the interneuron nature of the NB⁺Dx⁻ neurons. As an example, we presented NB⁺Dx⁻ V2a-INs (see *Supplementary Figure 13*). The exact nature (neurotransmitter phenotype and type) of the interconnected interneurons is an important question that we are currently working with. Similar to what we present in *Supplementary Figure 13*, we use a number of different transgenic lines, and we anticipate that in the near future, we will reveal the exact organization of the interconnected neurons and their dynamic modifications after injury (pathophysiological condition) or after exercise/training (physiological condition).

Figure 5 - for Reviewers. Confirmation that Dextran tracer labels the MN population in each spinal cord hemisegment.

8. L436: This sentence seems unnecessarily provocative. To me, it seems like past work has revealed the contribution of neurogenesis to regeneration, while your work reveals a novel mechanism that helps promote survival of existing neurons. I wouldn't describe your findings as 'arguing against' anything, but rather extending them in a complementary way.

Authors: We also received a similar comment from Reviewer#1 (see above response to Reviewer#1, minor comments #7). Although it might appear strong and potentially provocative at first, the rationale for this statement is the existing literature on zebrafish spinal cord injury and regeneration. According to the literature, the zebrafish's unique ability to regenerate its spinal cord is due to its innate capacity to generate new neurons, which is not the case for mammals. Our argument is that zebrafish can regenerate because it also employs different neuroprotective strategies and ensures that the neurogenesis process will build the physically damaged neurons. Hence, neuroprotection and neurogenesis are equally important for successful spinal cord regeneration. However, the role of neuroprotective mechanisms in the zebrafish is often overlooked in current literature as an equal or important contributor to the regeneration and restoration process. We anticipate that our studies will shift this view and bring both mechanisms under evaluation for potential use in mammalian spinal cord regeneration. After recognizing the strength of the original statement, we made an effort to tone down and soften our statements in the revised manuscript.

Reviewer #3:

Pedroni et al. have investigated cellular mechanisms that may support the remarkable ability of the adult zebrafish spinal cord to regenerate after complete spinal cord transection. They focused on the survival and sustained viability of large, rapidly-fatigable primary spinal motoneurons (pMNs) distal to the injury. As observed in studies of responses to spinal cord injury in other animals, they found that injury causes increases in extracellular glutamate that stimulate glutamatergic input to these neurons via AMPA and NMDA receptors, causes changes in their electrical excitability, and causes increase in their intracellular calcium following spinal cord stimulation. The increases in glutamatergic input and electrical excitability at 3dpi returned to control by 7dpi.

The level of parvalbumin remained constant in pMNs following spinal cord injury, but the level of calretinin increased by 1dpi and Calbindin mRNA (Calb2a) increased. Connexin Cx35/36 expression in pMNs and electrical coupling between pMNs also increased by 1dpi, peaked at 3dpi and declined by 7dpi. Injection of single pMNs with neurobiotin revealed dye coupling to increased numbers of other spinal neuron that peaked at 3dpi and returned to control numbers by 7dpi. As observed previously in rodents, the increases in connexin expression and electrical coupling depended on activation of mGluR2/3.

To test the bystander hypothesis that increased dye coupling of pMNs (CR+) to other neurons (CR-) buffers the calcium in the other neurons, the authors evaluated CR immunofluorescence intensity after treatment with a gap junction blocker carbenoxolone. CR intensity in pMNs was increased and the intensity of CR immunofluorescence was rescued in pMNs of injured fish. Neurons coupled to pMNs included v2a interneurons, suggesting that pMNs may rescue multiple populations of neurons. Treatment with carbenoxolone led to increased cell death of neurons with a range of sizes, assessed

by PI incorporation and bridging of rostral and caudal parts of the spinal cord was delayed at 7 and 14 dpi.

Overall this is a thorough investigation. However there are a few points that should be clarified.

Authors: We are thankful for the Reviewer's assessment of our work and the helpful comments that aim to improve our study substantially. We anticipate that this study will form the foundations for a more detailed dissection of the context-dependent neuronal spinal network transformations in the future. As such, some of the comments raised by the Reviewer are currently part of our ongoing studies.

Major concerns

1. It is puzzling to observe increased coupling between pMNs and other spinal neurons when connexin 35/36 is expressed only in pMNs. Are other connexins expressed in sMNs and other neurons? Are heterotypic gap junctions formed? What connexins are partnering with Cx35/36? Connexins only in pMNs would argue for hemichannels, lines 267-277 notwithstanding. This point needs to be resolved

Authors: We apologize if our previous text was unclear and led to any misunderstandings by the reviewer. In our study, we used an antibody against the mammalian Cx35/36 and found abundant labeling in the pMN somata, proximal dendrites, and proximal part of the axon. Please note that we never claimed that Cx35/36 is exclusively expressed in pMNs. We observed Cx35/36 in several neurons and sMNs, and it was also present in their somata (please refer to Supplementary Figure 7 of the previous version of the manuscript for more information).

Furthermore, we previously used the same antibody (Song et al., 2016) to study secondary MNs and V2a-INs. In those neurons, we observed that Cx35/36 is sparse in the soma area, and the vast expression exists in their dendrites and axons. Specifically, V2a-INs had higher labeling in their axon and axonal collaterals than in the dendrites, while secondary MNs showed a higher expression of Cx35/36 in their dendrites compared to their axons (Song et al., 2016; relative information displayed in Figure 1 on that article).

Whether the gap junction channels are homotypic or heterotypic still remains unclear, as it is largely neuron-to-neuron (or cell-to-cell) specific. For example, in zebrafish Mauthner cells, the gap junction channels are heterotypic, with the postsynaptic membrane having Cx35.5 and the presynaptic membrane having Cx34.1 (Martin et al., 2023).

However, in a recent study (Pallucchi et al., 2022) using the same non-commercial zebrafish-specific antibodies (Miller et al., 2017), the authors observed that pMNs had an abundance of Cx35.5 but no expression of the Cx34.1. Moreover, their results showed that the Cx35.5 contributes to the gap junctions between V2a-INs and pMNs, suggesting the presence of potentially homotypic channels in zebrafish spinal cord and particularly in pMNs and V2a-INs (Pallucchi et al., 2022).

It is important to note that we are currently uncertain whether the configuration we observed pertains to both uninjured and injured animals. Our analysis of the coupling coefficient indicated changes after injury (as shown in Figure 4m). In response to a comment from Reviewer#2 (please refer to Specific comments #1), we acknowledge that the gap junctional coupling and its configuration are more complex than we initially anticipated, and further studies in the future may help us resolve this issue. As such, we did not consider this information essential for our discussion before as we currently lack information on the topic and, therefore, did not include it in the previous version of the manuscript. However, the reviewer emphasized its significance, so we revised our manuscript to incorporate this information.

References:

- Martin EA, Michel JC, Kissinger JS, Echeverry FA, Lin YP, O'Brien J, Pereda AE, Miller AC. Neurobeachin controls the asymmetric subcellular distribution of electrical synapse proteins. *Curr Biol.* 2023 May 22;33(10):2063-2074.e4. doi: 10.1016/j.cub.2023.04.049. Epub 2023 May 11. PMID: 37172585; PMCID: PMC10266475.
- Miller AC, Whitebirch AC, Shah AN, Marsden KC, Granato M, O'Brien J, Moens CB. A genetic basis for molecular asymmetry at vertebrate electrical synapses. *Elife.* 2017 May 22;6:e25364. doi: 10.7554/eLife.25364. PMID: 28530549; PMCID: PMC5462537.
- Pallucchi I, Bertuzzi M, Michel JC, Miller AC, El Manira A. Transformation of an early-established motor circuit during maturation in zebrafish. *Cell Rep.* 2022 Apr 12;39(2):110654. doi: 10.1016/j.celrep.2022.110654. PMID: 35417694; PMCID: PMC9071512.
- Song J, Ampatzis K, Björnfors ER, El Manira A. Motor neurons control locomotor circuit function retrogradely via gap junctions. *Nature.* 2016 Jan 21;529(7586):399-402. doi: 10.1038/nature16497. Epub 2016 Jan 13. PMID: 26760208.

2. Does CR expression increase after injury in sMNs and other spinal neurons? Figure 3h,i and Supplementary Figure 6 suggest that it might be, but this is not explicitly stated.

Authors: In *Supplementary Figure 6* (the previous version of the manuscript is now *Supplementary Figure 8*), we show that spinal cord injury resulted in detecting 50% more spinal neurons that were CR⁺. However, our analysis did not include information regarding the nature of these neurons, interneurons, or secondary MNs. In our previous study (Berg et al., 2018), we found that the CR has differential expression within the secondary MNs. As such, ~84% of the fast sMNs express CR, ~17% of the intermediate sMNs, and none (0%) slow MN express CR. As such, and following the question raised by the Reviewer, we performed additional experiments to verify if there is any change regarding the type of secondary MNs that express CR after injury. Our experiments showed an increase in the proportion of sMNs that express CR, but no changes detected regarding CR expression intensity (see *Figure 6-for Reviewers*). Further analysis was done to detect if the increase was associated with a specific secondary MN pool. We detected that more Fast sMNs express CR at 3dpi (see *Figure 6-for Reviewers*). Now, these data are included in the revised supplementary Figure 8 (of the revised version of the paper) and in the revised manuscript text. The identification of all other neuronal types that start expressing CR after injury is an interesting and potentially informative question; however, it falls outside the scope of the current study. Yet, a recent study (Huang et al., 2022) showed that CR overexpressed in zebrafish spinal cord after injury, and the authors identified that after injury, more V2a-INs are CR⁺ confirming our studies that more neurons start expressing CR following a traumatic insult.

Figure 6- for Reviewers. Analysis of the CR expression in sMN population.

References:

- Berg EM, Bertuzzi M, Ampatzis K. Complementary expression of calcium binding proteins delineates the functional organization of the locomotor network. *Brain Struct Funct.* (2018) 223:2181-2196. doi: 10.1007/s00429-018-1622-4.
- Huang CX, Wang Z, Cheng J, Zhu Z, Guan NN, Song J. De novo establishment of circuit modules restores locomotion after spinal cord injury in adult zebrafish. *Cell Rep.* (2022) 41:111535. doi: 10.1016/j.celrep.2022.111535.

3. The increase in [Ca]_i in the other neurons (including V2a interneurons; these data were not included in Fig 3) needs to be quantified, since the test of the bystander hypothesis depends on elevation of [Ca]_i in these other neurons.

Authors: We agree that this was an important piece of information omitted from our previous version of the manuscript. We did this analysis (see *Figure 7-for Reviewers*), and we now included it in the manuscript as *Supplementary Figure 6*. After 3 days of injury, most neurons showed an increase in resting calcium, which was found to be unrelated to the size of the neurons.

Our data suggest a neuron-specific change in resting calcium after 3dpi because:

- The increase in the number of neurons (by 50%) expressing the calcium-binding protein Calretinin results in buffering the potential increase of calcium after an injury. This leads to the intracellular [Ca²⁺] being maintained at lower levels.
- The Bystander effect allows the spread of calcium through gap junction channels to multiple neurons. This phenomenon dampens the [Ca²⁺] between the interconnected population.
- General assessment has the potential to mislead. The Bystander effect requires the identification of the type of neuron as a donor or recipient of the calcium, Thus, a static observation of [Ca²⁺] in neurons is not informative *per se*.

In the presented study, the pMNs were the most accessible neurons for these types of analyses, as:

1. The number, size, and location of pMNs make them accessible for direct identification.
2. They have a high expression of Cx35/36 in their soma area, which is straightforward to evaluate.
3. They all express Calretinin and not stochastically after injury.

Finally, we only use the population of spinal interneurons called V2a-INs as an example to confirm that NB⁺ cells are indeed interneurons, as shown in *Supplementary Figure 13*. Currently, we are working on revealing the type of neurons connected by the bystander effect, which is an important question. However, these studies are challenging and require significant effort. We would like to have the time to conclude our experiment most solidly and appropriately. We anticipate that we will be able to share the results of these experiments with the rest of the community in the near future.

Figure 7- for Reviewers. Analysis of neuronal resting calcium in the adult zebrafish spinal cord.

4. The authors refer to “overuse” of the calretinin antibody (line 220) as the basis for a reduction in immunofluorescence of the antibody in Figure 3h. The conventional interpretation of immunofluorescence of an antibody is that it reflects the number of antigen binding sites and thus is proportional to the amount of the antigen (as in Supplementary Figure 6). Here it seems that the interpretation is not that the immunofluorescence refers to the amount of calretinin but to its ability to bind calcium. This would in principle depend on the binding site of the fluorophore and its proximity to the 3-D change in protein structure upon binding to calcium. Whether this Svant 6B3 antibody can report calcium independent of the amount of calretinin is not justified by citation or experiment in the manuscript. This point should be clarified. The continued presence of *Calb2a* mRNA is good to see, but does not in itself indicate the increased production of CR, since translation from mRNA to protein is a regulated step.

Authors: We are aware of the comment raised by the Reviewer. While we do not know in an absolute way if the epitopes and which ones are detected in the CR protein from the antibody used here, according to its 3D structure, we concluded that our antibody is most likely able to bind with higher affinity to the “free” (unsaturated) CR protein.

The reasons that make us adapt this model are the following:

1. Calretinin (CR) belongs to the so-called EF-hand family of calcium-binding proteins and, more specifically, is one of the most complicated proteins of the family as it contains 6 EF-Hand domains (Rogers, 1987; Schwaller, 2014). Similarly to many other proteins, the 3D structure can change significantly before and after binding to a selective molecule, in this case, calcium. Thus, the antibody could possibly not detect the epitopes after the structural changes of the protein.
2. The most **important reason** is that calretinin undergoes significant conformational changes after Ca²⁺ has become bound (Kuźnicki et al., 1995; Schwaller et al., 1997; Nikoletopoulou and Tavernarakis, 2012). These are not the usual protein 3D structural changes described above and are not reported for other members of the family, such as Parvalbumin and Calbindin (Nikoletopoulou and Tavernarakis, 2012). Accordingly, Ca²⁺-saturated calretinin is known to be cleaved between amino acids 60 and 61, yielding two main fragments. Moreover, these fragments will continue to be cleaved from the C-terminus end onwards (Kuźnicki et al., 1995). Based on this, we assume that our antibody will not be able to detect small fragments of CR protein after its saturation with calcium and during the cleaved conformational changes
3. Moreover, when we block the gap junction channels as we present in Figure 6, we show that this leads directly to a rescue of the calretinin intensity at 3 days post-injury (Figure 8 – for Reviewers). A simple intervention that fast and reliably restores the CR expression to the predicted high levels (supporting the *in situ* studies).
4. Finally, we agree that transcript levels (mRNAs) by themselves are not sufficient to predict protein concentration levels (Liu et al., 2016), and therefore, detection for *Calb2a* is not a determinant for calretinin protein expression *per se*. As such, previous studies into mRNA-protein correspondence have shown a poor correlation between mRNA and protein expression levels (Liu et al., 2016). Yet, exceptions to this general view are the differentially expressed mRNAs that correlate significantly stronger with their protein

product than the non-differentially expressed mRNAs (Koussounadis et al., 2015). Accordingly, a recent study in zebrafish demonstrated that *Calb2a* is differentially expressed in the adult spinal cord after injury (Huang et al., 2022), increasing our confidence that *Calb2a* expression to calretinin correspondence after injury is likely strong in this paradigm.

Collectively, all the data presented here and the current knowledge regarding the calretinin protein structure and changes suggest that the antibody staining probably has a higher affinity to calcium-free or less calcium-saturated calretinin. We want to assure the Reviewer that we had similar critical thinking about our claims regarding the “use of calretinin,” yet our data and literature forced us to adopt this notion. We have included some of this information in the revised manuscript.

Figure 8- for Reviewers. Calretinin intensity is rescued at 3dpi after blockage of GJC with CBX.

References:

- Huang CX, Wang Z, Cheng J, Zhu Z, Guan NN, Song J. De novo establishment of circuit modules restores locomotion after spinal cord injury in adult zebrafish. *Cell Rep.* (2022) 41:111535. doi: 10.1016/j.celrep.2022.111535.
- Koussounadis A, Langdon SP, Um IH, Harrison DJ, Smith VA. Relationship between differentially expressed mRNA and mRNA-protein correlations in a xenograft model system. *Sci Rep.* 2015 Jun 8;5:10775. doi: 10.1038/srep10775. PMID: 26053859; PMCID: PMC4459080.
- Kuźnicki J, Wang TL, Martin BM, Winsky L, Jacobowitz DM. Localization of Ca(2+)-dependent conformational changes of calretinin by limited tryptic proteolysis. *Biochem J.* 1995 Jun 1;308 (Pt 2)(Pt 2):607-12. doi: 10.1042/bj3080607. PMID: 7772048; PMCID: PMC1136969.
- Liu Y, Beyer A, Aebersold R. On the Dependency of Cellular Protein Levels on mRNA Abundance. *Cell.* 2016 Apr 21;165(3):535-50. doi: 10.1016/j.cell.2016.03.014. PMID: 27104977.
- Nikoletopoulou V, Tavernarakis N. Calcium homeostasis in aging neurons. *Front Genet.* 2012 Oct 2;3:200. doi: 10.3389/fgene.2012.00200. PMID: 23060904; PMCID: PMC3462315.
- Rogers JH. Calretinin: a gene for a novel calcium-binding protein expressed principally in neurons. *J Cell Biol.* 1987 Sep;105(3):1343-53. doi: 10.1083/jcb.105.3.1343. Erratum in: *J Cell Biol* 1990 May;110(5):1845. PMID: 3654755; PMCID: PMC2114790.
- Schwaller B, Durussel I, Jermann D, Herrmann B, Cox JA. Comparison of the Ca2+-binding properties of human recombinant calretinin-22k and calretinin. *J Biol Chem.* 1997 Nov 21;272(47):29663-71. doi: 10.1074/jbc.272.47.29663. PMID: 9368033.
- Schwaller B. Calretinin: from a "simple" Ca(2+) buffer to a multifunctional protein implicated in many biological processes. *Front Neuroanat.* 2014 Feb 5;8:3. doi: 10.3389/fnana.2014.00003. PMID: 24550787; PMCID: PMC3913827.

5. The authors place a good deal of trust in carbenoxolone as a specific blocker of electrical and dye coupling. Since specific blockade is central to the test of the bystander hypothesis, it would be helpful to know whether the major results are replicated with meclofenamic acid, a different allegedly specific blocker. It is gratifying that 200 µM CBX does not reveal PI+ cells. Death is probably the most extreme form of toxicity. Two blockers, each with different side effects or mild forms of toxicity, would make a more compelling case for the specificity of blockade.

Authors: The Reviewer raised a very good point regarding the gap junction blockers. Pharmacological agents that can act as potent gap junction blockers are widely available today. However, their precise mechanisms of action are not well-defined (Szarka et al., 2021). These agents are characterized by their non-specific nature, as they cannot differentiate between Cx hemichannels, GJs, or other membrane-bound channels (Manjarrez-Marmolejo and Franco-Pérez, 2016; Szarka et al., 2021). Despite this, they have been widely utilized and helped us to understand better the function of neuronal and non-neuronal gap junctions in several experimental settings and different animal model systems.

Our research required the use of a compound that would enable the pharmacological blockade of GJs. The efficacy of the GJ blockers has been principally evaluated using techniques to measure dye transfer (dye coupling) and electrical conductance (electrical coupling) (Abbaci et al., 2008); as such, we conducted tests on three different compounds (see below Figure 9 – for Reviewers). We tested the non-selective blocker Carbenoxolone and two more selective compounds, Mefloquine for 36, 43, and 50 isoforms of mammalian

connexins and Quinine for 36, 45, and 50 isoforms of mammalian connexins (Manjarrez-Marmolejo and Franco-Pérez, 2016; Szarka et al., 2021). We observed that Carbenoxolone and Mefloquine reduced electrical coupling between pMNs by 90.5% and 96.7%, respectively; they also significantly decreased the number of dye-coupled neurons. On the other hand, we observed that Quinine could reduce the electrical coupling between pMNs yet yield only a 50% reduction and could not effectively block the dye transfer. After analyzing our observations, we concluded that Carbenoxolone provided the best combination for blocking dye and electrical coupling.

Figure 9- for Reviewers. Evaluation of different blockers for gap junction channels in adult zebrafish spinal cord.

References:

Abbaci M, Barberi-Heyob M, Blondel W, Guillemain F, Didelon J. Advantages and limitations of commonly used methods to assay the molecular permeability of gap junctional intercellular communication. *Biotechniques*. 2008 Jul;45(1):33-52, 56-62. doi: 10.2144/000112810. PMID: 18611167.

Manjarrez-Marmolejo J, Franco-Pérez J. Gap Junction Blockers: An Overview of their Effects on Induced Seizures in Animal Models. *Curr Neuropharmacol*. 2016;14(7):759-71. doi: 10.2174/1570159x14666160603115942. PMID: 27262601; PMCID: PMC5050393.

Szarka G, Balogh M, Tengölics ÁJ, Ganczer A, Völgyi B, Kovács-Öller T. The role of gap junctions in cell death and neuromodulation in the retina. *Neural Regen Res*. 2021 Oct;16(10):1911-1920. doi: 10.4103/1673-5374.308069. PMID: 33642359; PMCID: PMC8343308.

Other points

Figure 1c. Please label the home segment (14?).

Authors: Done as suggested. The indication for the home segment has now been added.

Figure 2a. Please make the dashed lines white or change the figure legend.

Authors: Done as suggested.

Figure 3e,f. Where are l and j? Are they relics of previous lettering?

Authors: We apologize for that; we have now corrected it.

Figure 3g-i. What is μ about normalized intensity? Here and elsewhere, does μ signify “10-3”?

Authors: We apologize that we did not include the meaning of the μ meaning in the figure legends. The μ symbolizes in mathematics the population mean of pMNs belonging to the same spinal cord and, thus, to the same animal.

Lines 249,250. The coupling coefficient was not dependent on the distance between the neurons “over the distance tested (80 μm)” or “within a spinal segment”. The length constant is apparently long with respect to the length of a segment.

Authors: Thank you, we revised the text accordingly. The most distal pMNs within the same segment were approximately 80 μm apart. In our experimental young adult animals, the length of a spinal cord segment is approximately around 110-140 μm .

Figure 6a. Some white arrowheads do not seem to identify NB+CR- cell bodies.

Authors: Indeed, at first glance, it seems that there are some arrows pointing out not NB⁺CR⁻ neurons. However, this is one of the limitations of presenting whole-mount images from the adult zebrafish spinal cord. Below, we attached the same images (*Figure 10 – for Reviewers*), and we present alternative inserts for the channels NB and CR. In those examples, due to the tissue thickness (~50 μm) and weak NB labeling, the indicated neurons are not very visible. While this is one of the challenges in presenting our work, we would like to inform the Reviewer that all the evaluation of the NB⁺CR⁻ neurons, as well as other similar whole mount experiments, was performed using all the optical slices of the generated z-stack using both channels. Furthermore, the colocalization or not, of the signals on the same neurons was further evaluated using the orthogonal views generated in Fiji (ImageJ).

Figure 10 - for Reviewers. Presentation of different inserts with less obvious NB⁺CR⁻ neurons.

Supplementary Figure 7a. Are the + and - assignments inverted in the center panel of the bottom row?

Authors: We are not sure here about the image that the reviewers refer to. In case that is the one as pointed out below, then the symbols are correct. As the Reviewer can see in the magnification, the middle neuron has no expression of Cx35/36 in the soma and is therefore classified as (-). For the other 2 neurons on the right side, we could detect a few Cx35/36 punta on the soma; thus, we classified them as (+). We also checked all the images in the Figure to identify potential mistakes, but we could not.

Figure 11 - for Reviewers. Cx35/36 expression in neuronal somata (HuC/D+).

Supplementary Figure 10 title. It is not known whether CBX is without toxicity (adverse effects). It is shown that CBX affects pMN electrical and dye coupling without inducing PI uptake.

Authors: This is correct. We apologize for this wrong statement. We now revised the title accordingly.

Lines 351,352. ...“demonstrate that some of the cascade”....“vertebrates, the novel features have a profound”....

Authors: Thank you. We revised according to the suggestion.

REVIEWERS' COMMENTS

Reviewer #1 (Remarks to the Author):

I have read the authors' response to all reviewers' comments and the revised manuscript. I thank the authors for doing their best to address all raised comments, which certainly increased the support for their arguments as well as the clarity of their message.

I especially thank the authors for trying the ratio-metric calcium dye Fura-2, which seemed not to generate reliable labeling. In fact, effective labeling using synthetic calcium dyes depends very much on the tissue in which they are used, and even across brain regions, these dyes are known to generate various levels of labeling efficiency. Given the authors' reports on the lack of labeling efficiency of ratio-metric dyes, I am satisfied with the already presented results on Oregon-Green and Calcium Green dyes in the manuscript. I think the strongest results with these dyes are presented in Figures 3d-g. Especially the impact on longer calcium decay-time upon electrical stimulation clearly shows that calcium dynamics in these motor neurons are greatly altered. Moreover, the additional and newly presented analysis of calcium data is also in line with the authors' arguments and strongly supports the remaining part of the study with the impact of increased calcium on calcium-buffer-proteins such as Calretinin. Given all these, I appreciate the authors' efforts in addressing my comments on the presentation and interpretation of calcium imaging results to the best of their ability. I am satisfied with the presented revised results.

The electrophysiological results presented in Figure 2 with TTX and NMDA/AMPA blockers look very convincing.

I appreciate the additional discussion on the source of this glutamate surge after injury. I agree with the authors that the glia are likely to be the best candidate for this glutamate surge. This will likely become the point of focus in a future follow-up study.

The gap junction dye coupling results in Figure 4 looked highly convincing in the previous version of the manuscript. But I liked the revised Figure 4 now, upon request from Reviewer 2. I think that the authors' data looks very solid here.

I should finally state that the additional analysis and experiments presented for the nature and specificity of Calretinin and calcium increase upon the request of Reviewer 3 look highly convincing and interesting.

Given all this, I congratulate the authors on these exciting results. The revised manuscript supports their arguments and clearly communicates them. I fully support the publication of this manuscript in Nature Communications. I have no further comments.

Reviewer #2 (Remarks to the Author):

The authors have thoroughly addressed my comments.

Reviewer #3 (Remarks to the Author):

I thank the authors for their thoughtful and thorough responses to the initial review. Clarification of some of the presently unknown features of the system was particularly helpful (e.g. the neuron-specific identity of connexons) and the way in which calretinin is cleaved following calcium binding.

The manuscript and figures are clear and convincing. I'm glad to see the paper move forward!

Point-by-Point Response to Reviewers comments:**Reviewer #1:**

I have read the authors' response to all reviewers' comments and the revised manuscript. I thank the authors for doing their best to address all raised comments, which certainly increased the support for their arguments as well as the clarity of their message.

I especially thank the authors for trying the ratio-metric calcium dye Fura-2, which seemed not to generate reliable labeling. In fact, effective labeling using synthetic calcium dyes depends very much on the tissue in which they are used, and even across brain regions, these dyes are known to generate various levels of labeling efficiency. Given the authors' reports on the lack of labeling efficiency of ratio-metric dyes, I am satisfied with the already presented results on Oregon-Green and Calcium Green dyes in the manuscript. I think the strongest results with these dyes are presented in Figures 3d-g. Especially the impact on longer calcium decay-time upon electrical stimulation clearly shows that calcium dynamics in these motor neurons are greatly altered. Moreover, the additional and newly presented analysis of calcium data is also in line with the authors' arguments and strongly supports the remaining part of the study with the impact of increased calcium on calcium-buffer-proteins such as Calretinin. Given all these, I appreciate the authors' efforts in addressing my comments on the presentation and interpretation of calcium imaging results to the best of their ability. I am satisfied with the presented revised results.

The electrophysiological results presented in Figure 2 with TTX and NMDA/AMPA blockers look very convincing.

I appreciate the additional discussion on the source of this glutamate surge after injury. I agree with the authors that the glia are likely to be the best candidate for this glutamate surge. This will likely become the point of focus in a future follow-up study.

The gap junction dye coupling results in Figure 4 looked highly convincing in the previous version of the manuscript. But I liked the revised Figure 4 now, upon request from Reviewer 2. I think that the authors' data looks very solid here.

I should finally state that the additional analysis and experiments presented for the nature and specificity of Calretinin and calcium increase upon the request of Reviewer 3 look highly convincing and interesting.

Given all this, I congratulate the authors on these exciting results. The revised manuscript supports their arguments and clearly communicates them. I fully support the publication of this manuscript in Nature Communications. I have no further comments.

Authors: We appreciate the positive feedback and helpful suggestions from the reviewer. We acknowledge their comments, which aim to enhance the clarity and accessibility of our research and the reviewer's support for the publication of our work.

Reviewer #2:

The authors have thoroughly addressed my comments.

Authors: We are very pleased that we covered thoroughly and addressed adequately all of the Reviewer's comments.

Reviewer #3:

I thank the authors for their thoughtful and thorough responses to the initial review. Clarification of some of the presently unknown features of the system was particularly helpful (e.g. the neuron-specific identity of connexons) and the way in which calretinin is cleaved following calcium binding. The manuscript and figures are clear and convincing. I'm glad to see the paper move forward!

Authors: We wholeheartedly value the encouraging remarks and constructive feedback provided by the reviewer. We acknowledge and appreciate their invaluable suggestions, which effectively improve the presentation of our research and the Reviewer's unwavering support for our work to be published.